# THE THREE REGIMES OF OFFLINE-TO-ONLINE REIN-FORCEMENT LEARNING

## ABSTRACT

Offline-to-online reinforcement learning (RL) has emerged as a practical paradigm that leverages offline datasets for pretraining and online interactions for fine-tuning. However, its empirical behavior is highly inconsistent: design choices of online-fine tuning that work well in one setting can fail completely in another. We propose a stability–plasticity principle that can explain this inconsistency: we should preserve the knowledge of pretrained policy or offline dataset during online fine-tuning, whichever is better, while maintaining sufficient plasticity. This perspective identifies three regimes of online fine-tuning, each requiring distinct stability properties. We validate this framework through a large-scale empirical study, finding that the results strongly align with its predictions in 45 of 63 cases. This work provides a principled framework for guiding design choices in offline-to-online RL based on the relative performance of the offline dataset and the pretrained policy.

## 1 INTRODUCTION

Reinforcement learning (RL) has achieved impressive successes in a variety of domains (Mnih et al., 2015; Silver et al., 2017; Degrave et al., 2022), but its reliance on large amounts of online interaction often makes direct application to real-world problems challenging. To address this challenge, recent research has turned to leveraging pre-collected datasets for offline learning through offline RL (Levine et al., 2020) or imitation learning (Osa et al., 2018). This paradigm reduces the need for costly or unsafe online exploration by providing a strong initial policy trained entirely from offline data. However, policies trained purely offline are often suboptimal and fail to generalize to states outside the dataset's support, making online fine-tuning essential. Offline-to-online RL addresses this issue by first pretraining an agent on an offline dataset and then fine-tuning it with additional online interactions to further improve performance.

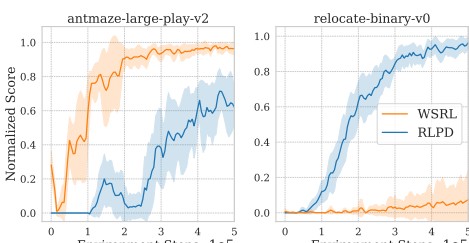

Figure 1: Comparison between **WSRL** (pretrained policy only) and **RLPD** (offline dataset only) on two representative offline-to-online RL tasks. All learning curves are shown as mean $\pm$ 95% CI.

While offline-to-online RL has led to promising results (Nair et al., 2020; Lee et al., 2022), online RL fine-tuning suffers from highly inconsistent empirical behavior: design choices that work well in one setting can fail completely in another. For example, as shown in Figure 1, on D4RL tasks (Fu et al., 2020) such as *antmaze-large-play-v2*, Warm-Start RL (WSRL) (Zhou et al., 2024), which relies on the pretrained policy and discards the offline dataset during online fine-tuning, substantially outperforms RL with Prior Data (RLPD) (Ball et al., 2023), which uses only the offline dataset. In contrast, on D4RL tasks such as *relocate-binary-v0*, the opposite pattern emerges, with RLPD outperforming WSRL by a wide margin. These seemingly inconsistent outcomes raise one fundamental question:

*What underlying factors cause design choices to succeed in some settings but fail in others?*

To answer this question, we propose a **stability–plasticity principle** for offline-to-online RL that can explain these seemingly inconsistent outcomes, which is inspired by prior research on plasticity and stability in neuroscience (McClelland et al., 1995) and machine learning (Kirkpatrick et al.,

Figure 2: **Overview of the three regimes in offline-to-online RL**, defined based on the *relative* performance of the pretrained policy $J(\pi_0)$ and the offline dataset $J(\pi_{\mathcal{D}})$. For each regime, our framework indicates which property is most needed during fine-tuning. The boxes at the right show representative design choices that implement these enhancing stability or plasticity. Dashed arrows denote weaker connections than solid arrows.

2017; Wolczyk et al., 2024; Dohare et al., 2024). Guided by the principle, effective fine-tuning requires a careful balance between stability and plasticity. **Stability** refers to the preservation of useful prior knowledge, ensuring that competencies acquired during pretraining are not substantially degraded. **Plasticity**, in contrast, denotes the capacity of the model to adapt flexibly and efficiently to new data. Furthermore, we identify two distinct forms of stability in offline-to-online RL: stability around the *pretrained policy* $\pi_0$, which emphasizes preserving knowledge explicitly encoded in the policy parameters, and stability around the *offline dataset* $\mathcal{D}$, which emphasizes retaining knowledge implicitly encoded in offline data. As stability and plasticity are inherently in trade-off, this distinction indicates that fine-tuning is more efficient when stability is enhanced with respect to the stronger source of prior knowledge, whether it is the pretrained policy or the offline dataset.

Building on this insight, we propose a taxonomy of **three regimes for offline-to-online RL**, each capturing a distinct relationship between the pretrained policy and the offline dataset. As shown in Figure 2, the three regimes are defined based on which source of prior knowledge is stronger, either the pretrained policy or the offline dataset. Moreover, different fine-tuning methods can be systematically categorized according to whether they enhance stability around $\pi_0$, stability around $\mathcal{D}$, or plasticity. By first determining the regime, one can select or design fine-tuning strategies that match its stability-plasticity requirements. This yields two *practical* benefits: it helps choose the most suitable method for each setting rather than applying a single uniform state-of-the-art algorithm, and it narrows the search space by indicating whether one should focus on leveraging the pretrained policy or on exploiting the offline dataset, thereby reducing unnecessary trial-and-error.

To validate this framework, we conduct a large-scale empirical study that covers 21 dataset-task compositions across four D4RL domains (MuJoCo locomotion, AntMaze, Adroit, and Kitchen) and three representative pretraining algorithms, yielding 63 settings. The results on stability, plasticity, and final performance, align closely with the predictions of our framework, supporting its utility as a principled basis for guiding design choices of fine-tuning in offline-to-online RL.

Contributions of this paper can be summarized as:

- We propose a stability–plasticity principle for offline-to-online RL that decomposes the fine-tuning performance into quantifiable stability and plasticity.
- Grounded in this principle, we develop a framework featuring a three-regime taxonomy defined by the relative performance of $\pi_0$ and $\mathcal{D}$.
- We conduct an extensive empirical study across 63 settings to validate and analyze this framework.

## 2 PRELIMINARY: OFFLINE-TO-ONLINE RL

Consider an MDP $\mathcal{M} = (\mathcal{S}, \mathcal{A}, P, R, \gamma)$, where the performance of a policy $\pi : \mathcal{S} \to \Delta(\mathcal{A})$ is measured by its expected discounted return: $J(\pi) = \mathbb{E}_{\pi,\mathcal{M}}[\sum_t \gamma^t r_t]$. Offline-to-online RL begins by pretraining the agent on an offline dataset $\mathcal{D}$, which is collected from $\mathcal{M}$ under an unknown behavior policy (or mixture of policies), using an offline RL algorithm $A_{\text{off}}$. This yields an offline pretrained agent whose policy is given by $\pi_0 = A_{\text{off}}(\mathcal{D})$. The fine-tuning step consists of using an online algorithm $A_{\text{on}}$ that starts from $\pi_0$ and interacts with $\mathcal{M}$ to obtain a final policy $\pi_N$ after $N$ iterations:

$$\pi_N = A_{\text{on}}(\mathcal{M}, \mathcal{D}, \pi_0). \tag{1}$$

The offline dataset $\mathcal{D}$ may be reused (Nakamoto et al., 2023) or discarded (Zhou et al., 2024) during fine-tuning. The objective of offline-to-online RL is to co-design $(A_{\text{off}}, A_{\text{on}})$ such that the final

policy $\pi_N$ maximizes $J(\pi_N)$. In this work, we focus on understanding and improving the **online RL fine-tuning** component $A_{\mathrm{on}}$ by fixing the offline pretraining component $A_{\mathrm{off}}$.

## 3 A STABILITY–PLASTICITY PRINCIPLE ON KNOWLEDGE ACQUISITION

This section introduces a formal framework for reasoning about fine-tuning in offline-to-online RL. Our goal is to characterize when and how online training leads to improvements over the offline initialization or degradations of what was already learned. We define two complementary properties of online fine-tuning: *stability*, the ability to preserve previously acquired performance, and *plasticity*, the capacity to improve further. These are grounded in the notion of a *knowledge level*, understood as the expected return encoded either in the dataset or in the pretrained policy. We show that final performance admits a decomposition into three terms — prior knowledge, stability, and plasticity. We leverage this perspective provides both theoretical insight and practical guidance.

### 3.1 QUANTIFYING PRIOR KNOWLEDGE

We distinguish two sources of prior knowledge available before online fine-tuning. The first is the knowledge encoded in the dataset, captured by the performance of the behavior policy that generated it. The second is the knowledge embodied in the pretrained policy, obtained by running an offline RL algorithm on the dataset. Both can serve as baselines for measuring stability.

**Knowledge from the dataset:** $J(\pi_{\mathcal{D}})$**.** Let $\mathcal{D}$ be the offline dataset and $\pi_{\mathcal{D}}$ be an abstract behavior policy representing the data-generating process. While $\mathcal{D}$ may be collected from a mixture of policies, $\pi_{\mathcal{D}}$ serves as a convenient abstraction. Its performance can be estimated by the average return:

$$J(\pi_{\mathcal{D}}) \approx \frac{1}{L} \sum_{k=1}^{L} \sum_{t=1}^{T} r_{k,t},$$

where $L$ is the number of trajectories in $\mathcal{D}$, and $r_{k,t}$ is the reward at time step $t$ in the $k$-th trajectory. This measure provides a scalar summary of the return level encoded in the dataset.[1]

**Knowledge from the pretrained policy:** $J(\pi_0)$**.** Let $\pi_0$ be the policy produced by applying an offline RL algorithm $A_{\mathrm{off}}$ to $\mathcal{D}$. Its performance, $J(\pi_0)$, reflects not only the quality of the dataset but also the inductive biases of $A_{\mathrm{off}}$ and the difficulty of the MDP $\mathcal{M}$. This policy can provide a strong initialization for online fine-tuning, though it does not necessarily dominate the dataset baseline. We therefore consider both $J(\pi_{\mathcal{D}})$ and $J(\pi_0)$ jointly as candidate sources of prior knowledge.

### 3.2 KNOWLEDGE DECOMPOSITION AND THE THREE REGIMES OF OFFLINE-TO-ONLINE RL

Online fine-tuning produces a sequence of policies $\{\pi_n\}_{n=0}^{N}$ with corresponding performances $\{J(\pi_n)\}_{n=0}^{N}$. Our goal is to understand how these trajectories of performance can be expressed in terms of the prior knowledge identified above and the two complementary properties of stability and plasticity. This leads to a decomposition of final performance that makes explicit what is preserved from offline training and what is gained during online interaction.

**Stability with regard to a knowledge level.** We define the stability of an online RL training process with respect to a knowledge level $l$, as the ability to retain the relative performance:

$$\mathrm{Stability}(l) := \min\left(\min_{0 \le n \le N} J(\pi_n) - l, 0\right). \tag{2}$$

This metric captures the worst-case *performance drop* during online fine-tuning relative to a given baseline $l$. A stability score of zero indicates that performance was never degraded below $l$; a negative score reflects how much was lost.

In our setting, the appropriate reference level is the best performance available from the offline pretraining phase, either from the dataset or from the pretrained policy:

$$J_{\mathrm{off}}^* := \max\left(J(\pi_0), \ J(\pi_{\mathcal{D}})\right). \tag{3}$$

---

[1] We do not assume access to additional quantities such as data coverage or performance surrogates (e.g., dense rewards for sparse-reward tasks), which could further enrich the quantification of prior knowledge.

We refer to this as the *offline performance baseline*, and the stability with respect to it is:

$$\text{Stability}(J_{\text{off}}^*) = \min_{0 \le n \le N} J(\pi_n) - J_{\text{off}}^* \le 0. \tag{4}$$

This condition ensures that the fine-tuning process does not degrade below the strongest available offline signal.

**Plasticity.** We define the plasticity as the ability to *acquire new knowledge* in online fine-tuning:

$$\text{Plasticity} := \max_{0 \le i \le N} J(\pi_i) - \min_{0 \le j \le N} J(\pi_j) \ge 0. \tag{5}$$

This metric measures the extent to which the performance can improve during fine-tuning, relative to its lowest observed value. A larger value indicates stronger capacity to learn from new data.

By relating these concepts through a **knowledge decomposition**, we have:

$$\underbrace{\max_{0 \le n \le N} J(\pi_n)}_{\text{Final knowledge}} = \underbrace{J_{\text{off}}^*}_{\text{Prior knowledge}} + \underbrace{\text{Stability}(J_{\text{off}}^*)}_{\text{Degradation on prior knowledge} \le 0} + \underbrace{\text{Plasticity}}_{\text{Online knowledge} \ge 0}. \tag{6}$$

This equation states that the final knowledge an agent acquires after offline pretraining and online fine-tuning is the outcome of three interacting components. (1) The first term is the prior knowledge provided by the offline phase, either through the dataset or the pretrained policy, whichever is stronger. (2) The second term measures stability, which records whether this prior knowledge is preserved or degraded during fine-tuning; it is always non-positive since performance can at best be maintained but not exceeded by this term. (3) The third term captures plasticity, the additional knowledge acquired through online interaction, which is non-negative by definition. Therefore, the *improvement* over the prior knowledge is given by the sum of plasticity and stability.

**The three regimes of offline-to-online RL.** With prior knowledge fixed, the objective of online fine-tuning is to improve final performance by maintaining stability with respect to $\max(J(\pi_0), J(\pi_\mathcal{D}))$ while ensuring sufficient plasticity. Based on this perspective, we identify three regimes for the online fine-tuning phase of offline-to-online RL: **Superior**: where $J(\pi_0) > J(\pi_\mathcal{D})$; **Comparable**: where $J(\pi_0) \approx J(\pi_\mathcal{D})$; **Inferior**: where $J(\pi_0) < J(\pi_\mathcal{D})$. These regimes are intended to reflect substantial differences in $J(\pi_0)$ and $J(\pi_\mathcal{D})$, since small performance gaps may not be meaningful and therefore should not determine regime assignment. We additionally explore alternative taxonomies using different metrics; the full details are given in Appendix C.

This regime taxonomy provides a principled framework for reasoning about the stability–plasticity trade-off in offline-to-online RL. It clarifies which source of prior knowledge should anchor stability in a given setting, as shown in Figure 2. In the **Superior** Regime, stability relative to $\pi_0$ should be prioritized, because $\pi_0$ contains more useful knowledge than $\mathcal{D}$. In the **Inferior** Regime, stability relative to $\mathcal{D}$ should be emphasized, as $\mathcal{D}$ contains more useful knowledge than $\pi_0$. In the **Comparable** Regime, both baselines provide similar knowledge, so preserving either helps. At the same time, maintaining sufficient plasticity across all regimes is essential for efficient fine-tuning. By categorizing design choices according to the stability or plasticity they promote, our framework turns what previously appeared as a disconnected set of practices into a structured and predictable landscape, providing a principled foundation for future algorithm design.

## 4 DESIGN CHOICES IN IMPROVING STABILITY OR PLASTICITY

Building on the stability–plasticity principle and the regime taxonomy, we now examine concrete design choices that instantiate these principles. To study these factors systematically, we isolate and analyze representative modules, each of which targets one of the three directions: stability around $\pi_0$, stability around $\mathcal{D}$, or increased plasticity. Although many existing offline-to-online RL algorithms combine multiple components that promote these aspects simultaneously, such entanglement makes it difficult to attribute effects to a specific source. To obtain a clearer scientific understanding, we isolate and analyze representative modules individually.

**Minimal baseline.** We begin with defining a naive online RL fine-tuning baseline that serves as the reference point for introducing additional components, which is *intentionally minimalist*. It

applies a standard online RL algorithm (e.g., SAC (Haarnoja et al., 2018)) initialized with an offline-pretrained agent, without any further modifications. This baseline anchors the analysis and makes the marginal effect of each added component interpretable.

## 4.1 STABILITY RELATIVE TO THE OFFLINE DATASET $\mathcal{D}$

Design choices in this category promote stability by reusing the offline dataset during online fine-tuning. Incorporating $\mathcal{D}$ into the online learning process helps preserve the knowledge in the offline dataset and mitigates distribution shift between offline and online data.

A common strategy for incorporating offline data during fine-tuning is to reuse the offline dataset together with newly collected online transitions. One approach initializes the replay buffer with the entire offline dataset, after which new online experiences are appended as the agent interacts with the environment. In this case, the ratio of offline to online data is determined by the dataset size and gradually shifts toward online data as training progresses. An alternative approach maintains two separate replay buffers: one fixed buffer containing the offline dataset and another buffer for online experiences. During training, each batch is sampled from both buffers according to a specified offline data ratio $\alpha$. For instance, RLPD (Ball et al., 2023) uses $\alpha = 0.5$, corresponding to a symmetric 50% offline and 50% online sampling ratio.

## 4.2 STABILITY WITH RESPECT TO THE PRETRAINED POLICY $\pi_0$

Design choices in this category focus on preserving and building upon the knowledge encoded in the pretrained policy $\pi_0$. The goal is to reduce the risk of catastrophic forgetting and ensure that fine-tuning does not erase useful behaviors learned during pretraining.

**Online data warmup.** Before applying gradient updates, the agent $\pi_0$ first collects a larger amount of online data ($K$ steps). This strategy reduces the mismatch between the pretraining distribution and the online data distribution, lowering the chance that early updates overwrite prior knowledge. It was introduced in WSRL (Zhou et al., 2024).

**Offline RL regularization.** Fine-tuning can also reuse the same offline RL algorithm that produced the pretrained policy, thereby inheriting its conservative regularization. This regularization penalizes state-action pairs outside the online data. Since the online data is collected by the sequence of policies from $\pi_0$ to $\pi_N$, the regularization *implicitly* anchors learning around the region visited by $\pi_0$, even if $\pi_0$ is not stored during fine-tuning. When offline data $\mathcal{D}$ is also used with regularization, we consider it as promoting stability towards *both* $\pi_0$ and $\mathcal{D}$; since $\pi_0$ is derived from $\mathcal{D}$, this setting tends to have the strongest stability. Such regularization is widely adopted in prior work (Nair et al., 2020; Kostrikov et al., 2021; Tarasov et al., 2023; Nakamoto et al., 2023).

## 4.3 PLASTICITY: PARAMETER RESET

A direct way to increase plasticity is to periodically reset network parameters while retaining the accumulated training data. Randomly initialized networks tend to exhibit higher plasticity than pretrained ones (Nikishin et al., 2022), although this strategy may come at the cost of severe forgetting and performance degradation. In the context of offline-to-online RL, parameter reset can be interpreted as starting from any pretrained agent $\pi_0$ and then resetting its weights to a randomly initialized agent $\pi_1$. While this approach severely degrades initial performance (i.e., $J(\pi_1) - J(\pi_0)$ is highly negative), it can significantly enhances plasticity. RLPD (Ball et al., 2023) directly trains an online RL agent from random initialization, which can be viewed as pretraining followed by fully parameter reset before fine-tuning.

## 5 EMPIRICAL STUDY

We test the validity of our three-regime framework through a large-scale empirical study, examining whether its regime-specific predictions align with observed outcomes. To connect the design modules described above with this framework, we group algorithms by the primary source of stability they emphasize. Methods that preserve knowledge from the pretrained policy $\pi_0$ are labeled $\boldsymbol{\pi_0}$-**centric**, while those that anchor stability to the offline dataset $\mathcal{D}$ are labeled $\boldsymbol{\mathcal{D}}$-**centric**. Approaches that combine elements of both are called **mixed $\boldsymbol{\pi_0 + \mathcal{D}}$ methods**. Finally, the **minimal baseline** corresponds to maximum plasticity with no explicit stability mechanism.

We study a diverse set of benchmark tasks and pretraining dataset compositions, following the experimental protocols of prior work (Nakamoto et al., 2023; Zhou et al., 2024). Specifically, we include MuJoCo locomotion, AntMaze, Kitchen, and Adroit domains from D4RL (Fu et al., 2020), covering a total of 21 dataset-task compositions. All experiments are conducted with 10 random seeds to ensure statistical reliability.

**Offline pretraining phase.** We employ two representative offline RL algorithms, Calibrated Q-Learning (CalQL) (Nakamoto et al., 2023) and ReBRAC (Tarasov et al., 2023), as well as behavior cloning (BC) (Schaal, 1996) using a deterministic policy. To pair a behavior-cloned policy with a critic, we pretrain the critic by FQE (Voloshin et al., 2019) after BC. Combining the 21 dataset-task compositions with these 3 pretraining algorithms yields 63 experimental settings in total. Each setting is defined by a specific combination of pretraining algorithm, dataset, and task. In this work, since we focus on cases where the offline dataset and the task are from the same MDP, the pretraining algorithm and dataset are sufficient to uniquely specify a setting.

We use the regime classification introduced in Section 3 to organize our analysis and to interpret the outcomes of the fine-tuning methods. Each of the 63 experimental settings, defined by a unique combination of pretraining algorithm, dataset, and task, is assigned to one of the three regimes based on the relative performance of the pretrained policy and the offline dataset. Specifically, we conduct $t$-tests with a margin $\delta = 0.05$ to assess whether the difference between $J(\pi_0)$ and $J(\pi_{\mathcal{D}})$ is statistically significant. The margin $\delta$ is introduced for robustness, since $J(\pi_{\mathcal{D}})$ is approximated by the dataset average return and small gaps between $J(\pi_0)$ and $J(\pi_{\mathcal{D}})$ may not be meaningful. It prevents over-interpreting numerical noise in regime assignment. The complete set of regime assignments is reported in Table 13 in the appendix.

**Online fine-tuning phase.** To ensure consistency between offline pretraining phase and online fine-tuning phase, we fine-tune each agent using the corresponding base algorithm. Specifically, we fine-tune CalQL-pretrained agents using SAC (Haarnoja et al., 2018) and ReBRAC-pretrained agents using TD3 (Fujimoto et al., 2018). For the deterministic BC pretraining, we use TD3 for fine-tuning to match its deterministic actor structure, and use ReBRAC when applying regularization.

Since evaluating every possible design and their combinations is infeasible, and we have grouped them into four categories: the minimal baseline, $\pi_0$-centric methods, $\mathcal{D}$-centric methods, and mixed $\pi_0 + \mathcal{D}$ methods. We then evaluate six representative methods spanning these four categories, which collectively capture the key design choices explored in prior offline-to-online RL literature.

- **Baseline:** Fine-tuning the pretrained policy using an online RL algorithm with only online data.
- **$\pi_0$-centric methods:** Two variants of such methods are evaluated: the baseline with (i) online data warmup ($K = 5{,}000$ steps) and (ii) offline RL regularization using the pretraining coefficient.
- **$\mathcal{D}$-centric methods:** Two variants of such methods are evaluated: the baseline with (i) offline data replay, and (ii) with offline data replay and reset. Both variants use separate replay buffers with an offline data ratio of $\alpha = 0.5$.
- **Mixed $\pi_0 + \mathcal{D}$ methods:** The baseline combined with offline data replay and offline RL regularization, a combination widely adopted in prior work (Nair et al., 2020; Kostrikov et al., 2021; Tarasov et al., 2023; Nakamoto et al., 2023).

In each setting, we focus and compare the strongest $\pi_0$-centric and $\mathcal{D}$-centric methods to better approximate the ideal performance achievable by each stability source, while minimizing confounding from implementation details and hyperparameter tuning. In the following subsections, we first present the empirical values of stability and plasticity across the three regimes and discuss how they align with intuition. We then present and analyze the fine-tuning results across the three regimes. Finally, we examine how the Q-function evolves during fine-tuning in the different regimes.

## 5.1 EMPIRICAL VALUES OF STABILITY AND PLASTICITY

To ground the concepts in knowledge decomposition (Eq. 6) in practice, we report the empirical values of stability, plasticity, and the improvement for different fine-tuning methods in Table 1 (**Superior** and **Inferior** regimes), Table 7 (**Comparable** regime), and Table 8 (all regimes).

We observe that the empirical values of stability and plasticity align well with intuition:

- Stability and Plasticity: The "offline RL + offline data" method attains the highest average stability across all regimes, as it improves stability relative to both $\pi_0$ and $\mathcal{D}$. It also achieves the highest

Table 1: **Empirical values of stability, plasticity, and improvement (stability + plasticity)** for different fine-tuning methods in the **Superior** and **Inferior** regimes (500k environment steps).

| Fine-tuning method | Stability ↑ | Plasticity ↑ | Improvement ↑ |
|---|---|---|---|
| **Superior Regime (500k environment steps)** | | | |
| baseline | $-0.399 \pm 0.363$ | $0.832 \pm 0.275$ | $0.433 \pm 0.276$ |
| + warmup ($\pi_0$-centric) | $-0.394 \pm 0.371$ | $0.865 \pm 0.251$ | $\mathbf{0.471 \pm 0.258}$ |
| + offline RL ($\pi_0$-centric) | $-0.199 \pm 0.268$ | $0.519 \pm 0.291$ | $0.319 \pm 0.256$ |
| + offline data ($\mathcal{D}$-centric) | $-0.376 \pm 0.380$ | $0.796 \pm 0.282$ | $0.421 \pm 0.251$ |
| + offline data + reset ($\mathcal{D}$-centric) | $-0.615 \pm 0.338$ | $\mathbf{1.011 \pm 0.183}$ | $0.396 \pm 0.271$ |
| + offline RL + offline data (mixed $\pi_0 + \mathcal{D}$) | $\mathbf{-0.124 \pm 0.213}$ | $0.393 \pm 0.288$ | $0.270 \pm 0.233$ |
| **Inferior Regime (500k environment steps)** | | | |
| baseline | $-0.692 \pm 0.307$ | $0.475 \pm 0.378$ | $-0.217 \pm 0.480$ |
| + warmup ($\pi_0$-centric) | $-0.677 \pm 0.306$ | $0.495 \pm 0.389$ | $-0.182 \pm 0.490$ |
| + offline RL ($\pi_0$-centric) | $-0.650 \pm 0.322$ | $0.262 \pm 0.283$ | $-0.388 \pm 0.489$ |
| + offline data ($\mathcal{D}$-centric) | $-0.674 \pm 0.308$ | $0.696 \pm 0.340$ | $0.023 \pm 0.451$ |
| + offline data + reset ($\mathcal{D}$-centric) | $-0.769 \pm 0.275$ | $\mathbf{0.870 \pm 0.299}$ | $\mathbf{0.101 \pm 0.205}$ |
| + offline RL + offline data (mixed $\pi_0 + \mathcal{D}$) | $\mathbf{-0.614 \pm 0.354}$ | $0.440 \pm 0.297$ | $-0.174 \pm 0.441$ |

Table 2: **Confusion matrix of fine-tuning results across the three regimes.** Green cells: correct predictions (45/63); red cells: opposite mismatches (3/63); gray cells: adjacent mismatches (15/63). Overall, the framework achieves 71% correct predictions with only 5% opposite mismatches.

| | | Pretraining Regime | | |
|---|---|---|---|---|
| | | **Superior** | **Comparable** | **Inferior** |
| **Fine-tune** | $\pi_0$-centric $>$ $\mathcal{D}$-centric | **24** | 2 | 1 |
| | $\pi_0$-centric $\approx$ $\mathcal{D}$-centric | 6 | **2** | 3 |
| | $\pi_0$-centric $<$ $\mathcal{D}$-centric | 2 | 4 | **19** |

plasticity in the **Comparable** regime. In contrast, "offline data + reset" yields the highest average plasticity in the **Superior** and **Inferior** regimes, but this comes at the cost of driving stability to its lowest possible level due to the full parameter reset. We also provide additional analysis in B.2.

- Improvement: The highest average improvement is regime-dependent. In the **Superior** regime, it is achieved by "warmup", which is a $\pi_0$-centric method; in the **Inferior** regime, it is achieved by "offline data + reset", which is a $\mathcal{D}$-centric method. In the **Comparable** regime, the best-performing method is "offline RL + offline data", with a small advantage. These patterns are consistent with intuition: in the **Superior** regime, $\pi_0$ provides more valuable knowledge to leverage, whereas in the **Inferior** regime, the offline dataset $\mathcal{D}$ contains the more valuable knowledge.

- Across versus within regimes: Aggregated results (Table 8) show modest differences across methods, but regime-specific tables reveal clear, method-dependent patterns, highlighting the practical utility of the stability–plasticity principle and the three-regime framework.

## 5.2 SUPERIOR REGIME: $J(\pi_0) > J(\pi_{\mathcal{D}})$

In this regime, the pretrained policy $\pi_0$ achieves substantially higher performance than the behavior policy, i.e., $J(\pi_0) > J(\pi_{\mathcal{D}})$. In such cases, the offline dataset offers limited additional value, and the primary concern becomes preserving stability relative to $\pi_0$ during online fine-tuning.

Representative results are shown in Figure 3, with the complete results in this regime provided in the appendix (Figure 7). We perform $t$-tests between the strongest $\pi_0$-centric and $\mathcal{D}$-centric methods in each setting. The results indicate $\pi_0$-centric methods outperform $\mathcal{D}$-centric methods in 24 out of 32 settings (75%), while the remaining settings mostly show no statistically significant difference. Only two settings exhibit the opposite outcome, and in both cases the difference in average performance is negligible. One such case is shown in the bottom-right plot of Figure 3. These statistics correspond to the **Superior** column of the confusion matrix in Table 2.

These aggregate outcomes strongly support our principle that, in the **Superior** regime, $\pi_0$-centric methods tend to be more effective than $\mathcal{D}$-centric methods. In other words, when the pretrained policy $\pi_0$ already outperforms the dataset (the superior case), methods that stick close to $\pi_0$ work better than those that keep leaning on the offline dataset. While the prediction accuracy is not perfect, such discrepancies are anticipated given the influence of hyperparameters and implementation details. Importantly, the overall observed patterns remain consistent with our principle.

Beyond aggregate comparisons, the analysis of specific design choices highlights key trade-offs between stability and plasticity. Within the two variants of $\pi_0$-centric methods, online data warmup

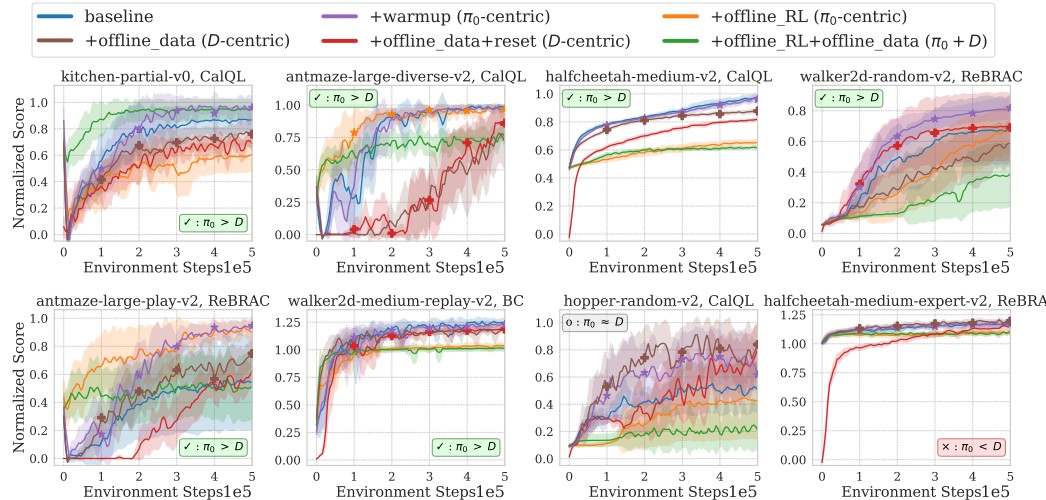

Figure 3: Representative fine-tuning results in the **Superior** regime: the first row and first two sub-plots in the second row are correct predictions, while the remaining two show an adjacent mismatch and an opposite mismatch. Markers on the curves indicate the better-performing variant within $\pi_0$-centric methods and within $\mathcal{D}$-centric methods.

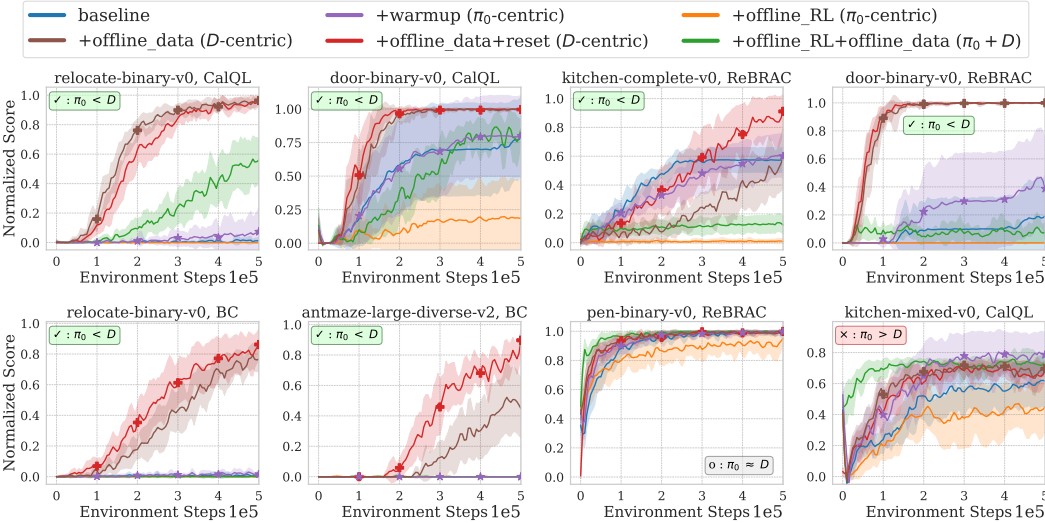

Figure 4: Representative results in the **Inferior** regime: the first six results are correct predictions, while the remaining two show an adjacent mismatch and an opposite mismatch.

achieves better performance in 27 out of 32 settings, reflecting its ability to preserve the pretrained policy's knowledge while maintaining sufficient plasticity for fine-tuning. In contrast, offline RL regularization provides stronger stability, leading to substantially less performance degradation during the early fine-tuning phase. However, this strong stability comes at the cost of reduced plasticity, which limits long-term improvements during fine-tuning, making it better than online data warmup in only 5 out of 32 settings, where the pretrained policy is already close to the optimal policy. The same applies to the combination of offline RL regularization with offline data replay, which exhibits the strongest stability among all methods considered. These contrasts highlight the importance of considering each setting and identifying the the method that best balances the underlying stability–plasticity trade-off.

> **Takeaway:** In the **Superior** regime, where the pretrained policy $\pi_0$ substantially outperforms the offline dataset $\mathcal{D}$, $\pi_0$-centric methods are typically more effective than $\mathcal{D}$-centric methods. Stronger stability proves beneficial primarily when $\pi_0$ is already close to optimal.

### 5.3 INFERIOR REGIME: $J(\pi_0) < J(\pi_{\mathcal{D}})$

In this regime, the pretrained policy $\pi_0$ performs much worse than the behavior policy underlying the offline dataset, i.e., $J(\pi_0) < J(\pi_{\mathcal{D}})$. Thus, $\pi_0$ contributes substantially less useful knowledge than the offline dataset, making it crucial to retain and leverage the offline data during online fine-tuning.

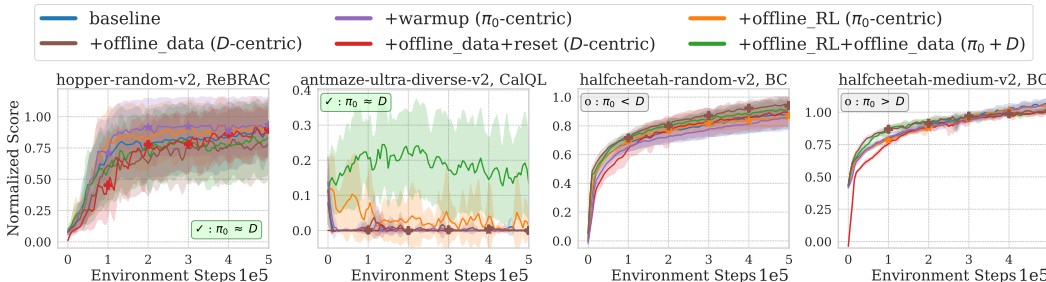

Figure 5: Representative fine-tuning results for the **Comparable** regime. The first two subplots illustrate cases consistent with our framework's predictions, while the latter two show mismatches with only small mean differences.

Representative results are shown in Figure 4, with the complete results in this regime provided in the appendix (Figure 8). $\mathcal{D}$-centric methods outperform $\pi_0$-centric methods in 19 out of 23 settings (83%), while the remaining settings mostly show no statistically significant difference. Only one setting exhibits the opposite outcome, which is illustrated in the rightmost plot of the second row in Figure 4. These statistics correspond to the **Inferior** column of the confusion matrix in Table 2. Taken together, these aggregate results show that in the **Inferior** regime, $\mathcal{D}$-centric methods tend to be more effective than $\pi_0$-centric methods, consistent with the prediction of our framework.

Notably, offline data replay with reset achieves better performance than offline data replay in 13 out of 24 settings, despite the fact that reset initially causes significant degradation. This indicates that in these cases the offline pretraining phase substantially reduces plasticity while offering limited useful knowledge, and resetting the parameters allows the agent to adapt and acquire new knowledge more effectively. Furthermore, combining offline RL regularization with offline data generally underperforms compared to $\mathcal{D}$-centric methods. Although this design leverages offline data during fine-tuning, which is essential in this regime, the excessive stability limits plasticity and thereby hinders further improvement.

**Takeaway:** In the **Inferior** regime, where the pretrained policy $\pi_0$ performs substantially worse than the offline dataset $\mathcal{D}$, $\mathcal{D}$-centric methods typically provide more effective fine-tuning than $\pi_0$-centric methods. Parameter reset can also be beneficial, particularly when the pretrained agent performs very poorly, as it restores plasticity and allows the agent to adapt more effectively.

### 5.4 COMPARABLE REGIME: $J(\pi_0) \approx J(\pi_{\mathcal{D}})$

In this regime, the pretrained policy and the behavior policy achieve similar performance, i.e., $J(\pi_0) \approx J(\pi_{\mathcal{D}})$. Representative results are shown in Figure 5, with the complete results provided in the appendix (Figure 9).

Our framework predicts that $\pi_0$-centric and $\mathcal{D}$-centric methods should yield comparable outcomes once fully optimized. Empirically, only 2 out of 8 settings are statistically indistinguishable under $t$-tests. This seems at odds with the prediction. However, closer inspection shows that the differences are minor: in 6 of 8 settings the mean gap between categories is less than 0.1. These small gaps indicate that both anchors provide similar prior knowledge, exactly as the framework suggests.

Why, then, do mismatches arise at all? The key is that effect sizes in this regime are small by construction. When $\pi_0$ and $\mathcal{D}$ are nearly tied, outcomes become highly sensitive to hyperparameters, initialization, and other implementation details. In our study, we fixed a limited set of representative variants and hyperparameters across all settings to avoid over-tuning. This conservative design choice helps comparability but can also tip results in such close cases.

**Takeaway:** In the **Comparable** regime, where the pretrained policy $\pi_0$ and the offline dataset $\mathcal{D}$ exhibit similar performance, $\pi_0$-centric and $\mathcal{D}$-centric methods should in principle yield comparable fine-tuning outcomes when fully optimized, though in practice their relative performance is often sensitive to implementation details.

### 5.5 ANALYSIS OF Q-FUNCTION DURING FINE-TUNING

Lastly, because critic learning is central to value-based methods, we examine how the Q-function evolves during fine-tuning in the **Superior** and **Inferior** regimes. Figure 6 illustrates a representative

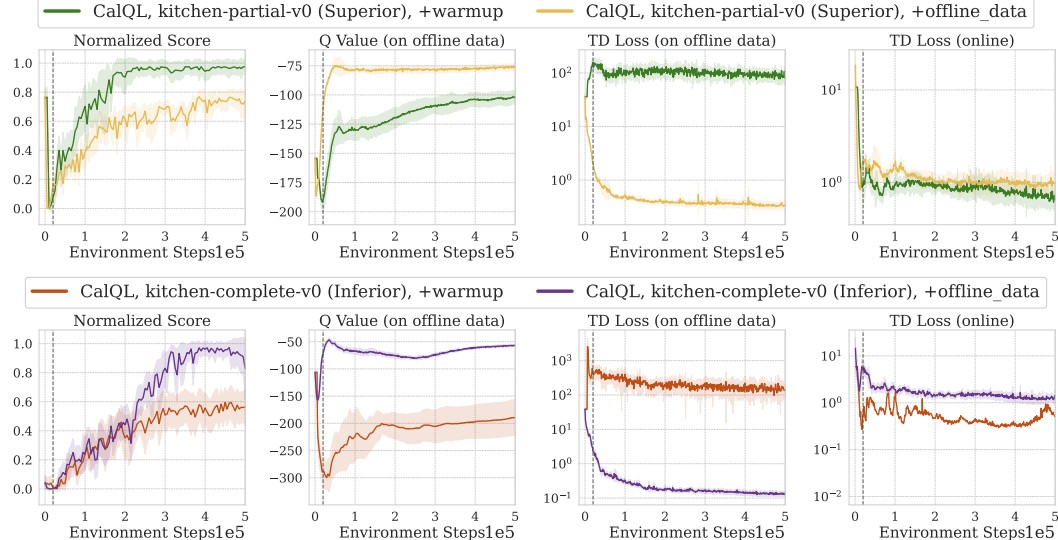

Figure 6: Comparison of Q-function behavior in different regimes. Fine-tuning without offline data results in significantly higher TD loss on the offline dataset in the **Inferior** regime compared to the **Superior** regime, and also leads to divergence of the Q-values on the offline dataset.

pattern using CalQL pretraining, with one **Superior** regime (*kitchen-partial-v0*) and one **Inferior** regime (*kitchen-complete-v0*). In each regime, we compare a $\pi_0$-centric method (warmup) with a $\mathcal{D}$-centric method (offline data).

In both regimes, the two methods exhibit similarly low TD loss on the online data distribution. As expected, fine-tuning with the offline dataset results in small TD loss on the offline dataset. However, warmup (without offline dataset) produces much larger TD loss on the offline dataset in the **Inferior** regime than in the **Superior** regime, especially in the first 100k steps, and the corresponding Q-values diverge more severely. This further illustrates why the offline dataset is crucial in the **Inferior** regime, whereas it is not necessary in the **Superior** regime.

## 6 CONCLUSION

This paper introduced the stability–plasticity principle as a way to reconcile the puzzling variability of offline-to-online RL. We showed that the key determinant of fine-tuning success is which source of prior knowledge (pretrained policy or offline dataset—serves as the stronger anchor). From this observation we derived a taxonomy of three regimes, each dictating where stability should be enforced and how plasticity should be managed. The value of this framework is twofold. First, it provides a clear explanation for the conflicting empirical evidence in the literature: design choices that seem inconsistent across benchmarks in fact reflect different underlying regimes. Second, it offers actionable guidance for practitioners. By identifying the regime of a given setting, one can select methods that align with its stability–plasticity requirements, reducing reliance on trial-and-error.

Our regime taxonomy provides an efficient lens for understanding offline-to-online RL by compressing complex phenomena into a small number of discrete categories. Such taxonomies have been highly successful in both the natural sciences and machine learning, for example in imitation learning, where regimes have been proposed based on density ratios (Spencer et al., 2021) or dataset size (Belkhale et al., 2023). At the same time, we recognize that discretizing behavior into three regimes is a simplification, particularly in situations where return does not adequately capture the usefulness of pretrained policies or offline datasets, for example in long-horizon sparse-reward settings. Real systems often lie along a continuum, and other dataset characteristics, such as coverage, play an important role but remain difficult to capture consistently. Extending the framework to incorporate such dimensions is an important direction for future work.

Our work also connects offline-to-online RL to a broader body of research on the stability–plasticity dilemma in deep learning and neuroscience. Recent studies in continual and online deep RL have focused on characterizing forgetting and plasticity, but applications to the offline-to-online transition have been limited. We show that stability–plasticity is not only a useful lens for analyzing this setting, but also a source of practical guidance: it predicts which design choices are effective in which regimes, moving beyond coining terminology to actionable prescriptions.

ETHICS STATEMENT

All authors have read and adhered to the ICLR Code of Ethics.

REPRODUCIBILITY STATEMENT

We provide the complete source code in the supplementary material and a comprehensive description of the experimental setup in the Appendix.

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

## A  RELATED WORK

**Offline-to-online RL.**  Offline-to-online RL seeks to combine the strengths of offline pretraining with the adaptability of online fine-tuning. Early approaches focused on extending offline RL regularization methods into the online regime, constraining fine-tuning updates to remain close to the pretrained policy. For example, Advantage Weighted Actor-Critic (AWAC) (Nair et al., 2020) and Implicit Q-Learning (IQL) (Kostrikov et al., 2021) applied offline regularization techniques directly to online fine-tuning. Building on the similar idea, PROTO (Li et al., 2023) introduced KL regularization to explicitly constrain the online policy to the pretrained one. Another line of work proposes new replay strategies for incorporating offline data more effectively. Lee et al. (2022) propose balanced replay to mitigate distribution shift and bootstrap error when transitioning from offline to online learning, Liu et al. (2024) employs a diffusion model to select or generate samples. Alternative strategies separate the roles of exploration and exploitation during fine-tuning. Jump-Start RL (JSRL) (Uchendu et al., 2023), maintains a fixed guided policy from pretraining alongside an exploration policy that is updated online, progressively transferring control from the pretrained policy to the learned one. More recent directions include expanding the action space via policy set expansion (PEX (Zhang et al., 2023)) and Bayesian methods for uncertainty-aware exploration (BOORL (Hu et al., 2024)). Some approaches skip offline pretraining but still make use of offline data. For example, Song et al. (2022) and Ball et al. (2023) start training directly with online RL while incorporating offline datasets, providing another way to combine offline data with online interaction.

**Plasticity and Stability.**  Plasticity and stability have long been recognized as central, often competing, objectives in learning systems. In neuroscience, this tension is formalized as the stability–plasticity dilemma (Mermillod et al., 2013), highlighting the challenge of integrating new knowledge without overwriting previously acquired competencies. In machine learning, similar dynamics manifest when agents must adapt to new data while preserving useful prior knowledge. Early work on continual and lifelong learning addressed this challenge via regularization techniques (Kirkpatrick et al., 2017), replay buffers (Rolnick et al., 2019) and modular architectures (Rusu et al., 2016). More recently, researchers have observed that insufficient plasticity can also hinder online deep RL, motivating methods designed to enhance plasticity of the neural network during training (Nikishin et al., 2022; Sokar et al., 2023; Dohare et al., 2024). In our work, we extend this perspective by framing offline-to-online RL explicitly as a stability–plasticity problem, where the central challenge is to balance the preservation of prior knowledge from pretraining with the adaptability needed to learn from new online experiences.

## B  DETAILED EXPERIMENTAL SETUP AND COMPLETE RESULTS

### B.1  OFFLINE PRETRAINING

For offline pretraining, we train CalQL for 1M gradient steps on AntMaze, 20k on Adroit, and 250k on both Kitchen and MuJoCo locomotion tasks. ReBRAC is trained for 1M gradient steps on AntMaze, 100k on Adroit, 250k on Kitchen, and 500k on MuJoCo tasks. For the behavior cloning (BC) baseline, we perform 500K gradient steps of policy learning followed by 100k steps of fitted

Q evaluation (Voloshin et al., 2019) (FQE) to obtain a Q-function for subsequent RL fine-tuning. Since this work primarily focuses on the online fine-tuning stage of offline-to-online RL, we do not modify the offline pretraining algorithm or its default hyperparameters.

For regime classification, we employ the two one-sided $t$-test (TOST) procedure with a margin of $\delta = 0.05$ and a significance level of $\alpha = 0.05$. The goal is to formally assess whether the pretrained policy $\pi_0$ and the offline dataset $\pi_{\mathcal{D}}$ are statistically indistinguishable in performance, or whether one is significantly superior. Let $\mu_0$ and $\mu_{\mathcal{D}}$ denote the mean returns of $\pi_0$ and $\pi_{\mathcal{D}}$. We conduct two one-sided tests for the null hypotheses $H_0 : \mu_0 - \mu_{\mathcal{D}} \leq -\delta$ and $H_0 : \mu_0 - \mu_{\mathcal{D}} \geq \delta$. If both null hypotheses are rejected, the difference is within the margin and the two are considered comparable, leading to assignment to the **Comparable** Regime. If only one hypothesis is rejected, the difference is statistically significant and exceeds the margin, and the setting is assigned to either the **Superior** or **Inferior** Regime depending on which policy achieves the higher mean return. The statistics for each dataset and pretraining policy are reported in Table 13.

While we use the margin parameter $\delta = 0.05$ in the main experiments, we further assess the sensitivity to $\delta$ by comparing results under $\delta = 0$ and $\delta = 0.1$. The corresponding fine-tuning confusion matrices are reported in Table 3 and Table 4.

Table 3: Confusion matrix of fine-tuning results across the three pretraining regimes with margin $\delta = 0$.

|  |  | Pretraining Regime | | |
| --- | --- | --- | --- | --- |
|  |  | **Superior** | **Comparable** | **Inferior** |
| **Fine-tune** | $\pi_0$-centric $> \mathcal{D}$-centric | **26** | 0 | 1 |
|  | $\pi_0$-centric $\approx \mathcal{D}$-centric | 8 | **0** | 3 |
|  | $\pi_0$-centric $< \mathcal{D}$-centric | 4 | 1 | **20** |

Table 4: Confusion matrix of fine-tuning results across the three pretraining regimes with margin $\delta = 0.1$.

|  |  | Pretraining Regime | | |
| --- | --- | --- | --- | --- |
|  |  | **Superior** | **Comparable** | **Inferior** |
| **Fine-tune** | $\pi_0$-centric $> \mathcal{D}$-centric | **18** | 9 | 0 |
|  | $\pi_0$-centric $\approx \mathcal{D}$-centric | 4 | **5** | 2 |
|  | $\pi_0$-centric $< \mathcal{D}$-centric | 2 | 9 | **14** |

## B.2 ONLINE FINE-TUNING

Across all environments, online fine-tuning is performed for 500k environment steps with UTD=1. The complete results, categorized according to the regime taxonomy, are reported in Figure 7, Figure 8, and Figure 9. To obtain the strongest performance for each class in each settings, we compare the interquartile mean (IQM) of evaluation results of online data warmup and offline RL regularization within $\pi_0$-centric methods, and analogously compare offline data replay with and without reset within $\mathcal{D}$-centric methods. We then use the higher value from each class, comparing $\pi_0$-centric and $\mathcal{D}$-centric methods using two-sided $t$-tests with $\alpha = 0.05$. To obtain stable and reliable $t$-test statistics, we base our analysis on the last 10 evaluation results from each random seed during online fine-tuning, which correspond to the final 50k training steps given our evaluation frequency of every 5k steps. An exception is made for *door-binary-v0* and *pen-binary-v0*, where we instead use results up to 200k steps, since by the end of training nearly all methods achieve a 100% success rate, leaving no differences.

We report the empirical values of stability, plasticity, and improvement during fine-tuning for the **Superior**, **Inferior**, **Comparable**, and all regimes. Since these quantities depend on the fine-tuning steps, we present results at both 50k and 500k environment steps, representing the *early* and *late* stages of fine-tuning, as shown in Table 5–Table 8.

**Further discussion on early and late fine-tuning stages.** Plasticity often correlates with performance improvement, but it is not sufficient on its own to guarantee strong results. Our analyses reveal the following regime- and stage-dependent patterns:

- **Superior regime.** The "offline data + reset" method attains the highest plasticity, yet yields the *lowest* improvement at 50k steps and only moderate improvement at 500k steps. This shows that when $\pi_0$ is strong, plasticity alone does not determine performance; maintaining stability is essential.
- **Inferior regime.** The relationship between stability, plasticity, and improvement depends on the fine-tuning stage. At the late stage (500k steps), methods with the highest plasticity tend to achieve the greatest improvement because $\pi_0$ performs poorly and can degrade toward near-zero performance during fine-tuning. In cases where $\min_j J(\pi_j) = 0$, the improvement simplifies to plasticity $- J_{\text{off}}^*$, making plasticity the dominant factor. In contrast, at the early stage (50k steps), the "offline RL + offline data" method attains the highest improvement while also exhibiting the strongest stability, despite having only moderate plasticity. This indicates that stability can have a stronger influence on improvement early in fine-tuning.

**Takeaway:** Plasticity should be emphasized when $\pi_0$ is weak (**Inferior** regime) and the fine-tuning budget is large (500k steps), while stability becomes more important when $\pi_0$ is strong (**Superior** regime) and the fine-tuning budget is small (50k steps).

Table 5: Empirical values of stability, plasticity, and improvement of different fine-tuning methods in **Superior** Regime for 50k and 500k environment steps.

| Fine-tuning method in **Superior** regime | Stability ↑ | Plasticity ↑ | Improvement ↑ |
|---|---|---|---|
| **50k environment steps** | | | |
| baseline | $-0.352 \pm 0.344$ | $0.525 \pm 0.325$ | $0.172 \pm 0.145$ |
| + warmup ($\pi_0$-centric) | $-0.328 \pm 0.348$ | $0.501 \pm 0.315$ | $\mathbf{0.173 \pm 0.122}$ |
| + offline RL ($\pi_0$-centric) | $-0.162 \pm 0.242$ | $0.289 \pm 0.255$ | $0.127 \pm 0.151$ |
| + offline data ($\mathcal{D}$-centric) | $-0.293 \pm 0.351$ | $0.448 \pm 0.318$ | $0.155 \pm 0.138$ |
| + offline data + reset ($\mathcal{D}$-centric) | $-0.615 \pm 0.338$ | $\mathbf{0.581 \pm 0.349}$ | $-0.034 \pm 0.311$ |
| + offline RL + offline data (mixed $\pi_0 + \mathcal{D}$) | $\mathbf{-0.072 \pm 0.128}$ | $0.204 \pm 0.189$ | $0.132 \pm 0.138$ |
| **500k environment steps** | | | |
| baseline | $-0.399 \pm 0.363$ | $0.832 \pm 0.275$ | $0.433 \pm 0.276$ |
| + warmup ($\pi_0$-centric) | $-0.394 \pm 0.371$ | $0.865 \pm 0.251$ | $\mathbf{0.471 \pm 0.258}$ |
| + offline RL ($\pi_0$-centric) | $-0.199 \pm 0.268$ | $0.519 \pm 0.291$ | $0.319 \pm 0.256$ |
| + offline data ($\mathcal{D}$-centric) | $-0.376 \pm 0.380$ | $0.796 \pm 0.282$ | $0.421 \pm 0.251$ |
| + offline data + reset ($\mathcal{D}$-centric) | $-0.615 \pm 0.338$ | $\mathbf{1.011 \pm 0.183}$ | $0.396 \pm 0.271$ |
| + offline RL + offline data (mixed $\pi_0 + \mathcal{D}$) | $\mathbf{-0.124 \pm 0.213}$ | $0.393 \pm 0.288$ | $0.270 \pm 0.233$ |

### B.3 HYPERPARAMETERS

We summarize the hyperparameters used in our empirical studies. The hyperparameters for SAC and CalQL are given in Table 9. Common hyperparameters of TD3 and ReBRAC appear in Table 10, while Table 11 contains the task-dependent hyperparameters for ReBRAC. Table 12 reports the hyperparameters for BC and FQE.

### B.4 COMPUTE DETAILS

All experiments were conducted on a single-GPU setup using an NVIDIA L40S GPU, 24 CPU workers, and 20GB of RAM.

Table 6: Empirical values of stability, plasticity and improvement of different fine-tuning methods in **Inferior** Regime for 50k and 500k environment steps.

| Fine-tuning method in **Inferior** regime | Stability ↑ | Plasticity ↑ | Improvement ↑ |
|---|---|---|---|
| **50k environment steps** | | | |
| baseline | $-0.688 \pm 0.348$ | $0.216 \pm 0.219$ | $-0.472 \pm 0.408$ |
| + warmup ($\pi_0$-centric) | $-0.671 \pm 0.343$ | $0.213 \pm 0.217$ | $-0.458 \pm 0.422$ |
| + offline RL ($\pi_0$-centric) | $-0.665 \pm 0.342$ | $0.159 \pm 0.197$ | $-0.505 \pm 0.428$ |
| + offline data ($\mathcal{D}$-centric) | $-0.670 \pm 0.343$ | $0.268 \pm 0.215$ | $-0.402 \pm 0.398$ |
| + offline data + reset ($\mathcal{D}$-centric) | $-0.832 \pm 0.191$ | $\mathbf{0.417 \pm 0.301}$ | $-0.415 \pm 0.395$ |
| + offline RL + offline data (mixed $\pi_0 + \mathcal{D}$) | $\mathbf{-0.640 \pm 0.357}$ | $0.266 \pm 0.219$ | $\mathbf{-0.374 \pm 0.415}$ |
| **500k environment steps** | | | |
| baseline | $-0.692 \pm 0.307$ | $0.475 \pm 0.378$ | $-0.217 \pm 0.480$ |
| + warmup ($\pi_0$-centric) | $-0.677 \pm 0.306$ | $0.495 \pm 0.389$ | $-0.182 \pm 0.490$ |
| + offline RL ($\pi_0$-centric) | $-0.650 \pm 0.322$ | $0.262 \pm 0.283$ | $-0.388 \pm 0.489$ |
| + offline data ($\mathcal{D}$-centric) | $-0.674 \pm 0.308$ | $0.696 \pm 0.340$ | $0.023 \pm 0.451$ |
| + offline data + reset ($\mathcal{D}$-centric) | $-0.769 \pm 0.275$ | $\mathbf{0.870 \pm 0.299}$ | $\mathbf{0.101 \pm 0.205}$ |
| + offline RL + offline data (mixed $\pi_0 + \mathcal{D}$) | $\mathbf{-0.614 \pm 0.354}$ | $0.440 \pm 0.297$ | $-0.174 \pm 0.441$ |

Table 7: Empirical values of stability, plasticity and improvement of different fine-tuning methods in **Comparable** Regime for 50k and 500k environment steps.

| Fine-tuning method in **Comparable** regime | Stability ↑ | Plasticity ↑ | Improvement ↑ |
|---|---|---|---|
| **50k environment steps** | | | |
| baseline | $-0.083 \pm 0.084$ | $0.163 \pm 0.237$ | $0.080 \pm 0.251$ |
| + warmup ($\pi_0$-centric) | $-0.082 \pm 0.085$ | $0.163 \pm 0.195$ | $0.081 \pm 0.212$ |
| + offline RL ($\pi_0$-centric) | $-0.079 \pm 0.085$ | $0.214 \pm 0.264$ | $0.135 \pm 0.270$ |
| + offline data ($\mathcal{D}$-centric) | $-0.081 \pm 0.085$ | $0.147 \pm 0.161$ | $0.066 \pm 0.166$ |
| + offline data + reset ($\mathcal{D}$-centric) | $-0.114 \pm 0.147$ | $\mathbf{0.261 \pm 0.250}$ | $0.147 \pm 0.237$ |
| + offline RL + offline data (mixed $\pi_0 + \mathcal{D}$) | $\mathbf{-0.057 \pm 0.058}$ | $0.240 \pm 0.251$ | $\mathbf{0.182 \pm 0.258}$ |
| **500k environment steps** | | | |
| baseline | $-0.043 \pm 0.071$ | $0.560 \pm 0.411$ | $0.516 \pm 0.452$ |
| + warmup ($\pi_0$-centric) | $-0.042 \pm 0.072$ | $0.543 \pm 0.390$ | $0.501 \pm 0.423$ |
| + offline RL ($\pi_0$-centric) | $-0.043 \pm 0.072$ | $0.564 \pm 0.382$ | $0.521 \pm 0.416$ |
| + offline data ($\mathcal{D}$-centric) | $-0.042 \pm 0.072$ | $0.680 \pm 0.338$ | $0.638 \pm 0.361$ |
| + offline data + reset ($\mathcal{D}$-centric) | $-0.106 \pm 0.139$ | $0.623 \pm 0.431$ | $0.516 \pm 0.431$ |
| + offline RL + offline data (mixed $\pi_0 + \mathcal{D}$) | $\mathbf{-0.035 \pm 0.061}$ | $\mathbf{0.682 \pm 0.302}$ | $\mathbf{0.646 \pm 0.314}$ |

Table 8: Empirical values of stability, plasticity and improvement of different fine-tuning methods across all regimes for 50k and 500k environment steps.

| Fine-tuning method | Stability ↑ | Plasticity ↑ | Improvement ↑ |
|---|---|---|---|
| **50k environment steps** | | | |
| baseline | $-0.433 \pm 0.379$ | $0.397 \pm 0.329$ | $-0.036 \pm 0.395$ |
| + warmup ($\pi_0$-centric) | $-0.413 \pm 0.380$ | $0.382 \pm 0.315$ | $-0.031 \pm 0.389$ |
| + offline RL ($\pi_0$-centric) | $-0.311 \pm 0.359$ | $0.242 \pm 0.246$ | $-0.069 \pm 0.404$ |
| + offline data ($\mathcal{D}$-centric) | $-0.391 \pm 0.387$ | $0.365 \pm 0.298$ | $-0.026 \pm 0.357$ |
| + offline data + reset ($\mathcal{D}$-centric) | $-0.632 \pm 0.349$ | $0.486 \pm 0.340$ | $-0.146 \pm 0.393$ |
| + offline RL + offline data (mixed $\pi_0 + \mathcal{D}$) | $-0.247 \pm 0.345$ | $0.226 \pm 0.206$ | $-0.021 \pm 0.357$ |
| **500k environment steps** | | | |
| baseline | $-0.461 \pm 0.381$ | $0.667 \pm 0.375$ | $0.206 \pm 0.502$ |
| + warmup ($\pi_0$-centric) | $-0.453 \pm 0.382$ | $0.689 \pm 0.372$ | $0.236 \pm 0.495$ |
| + offline RL ($\pi_0$-centric) | $-0.344 \pm 0.362$ | $0.431 \pm 0.327$ | $0.087 \pm 0.526$ |
| + offline data ($\mathcal{D}$-centric) | $-0.442 \pm 0.388$ | $0.745 \pm 0.316$ | $0.303 \pm 0.415$ |
| + offline data + reset ($\mathcal{D}$-centric) | $-0.607 \pm 0.360$ | $0.910 \pm 0.299$ | $0.304 \pm 0.318$ |
| + offline RL + offline data (mixed $\pi_0 + \mathcal{D}$) | $-0.291 \pm 0.360$ | $0.447 \pm 0.307$ | $0.155 \pm 0.433$ |

Table 9: SAC and CalQL's hyperparameters.

| Parameter | Value |
|---|---|
| optimizer | Adam |
| batch size | 1024 |
| learning rate | 1e-4 |
| Q-function soft-update rate ($\tau$) | 5e-3 |
| discount factor ($\gamma$) | 0.999 on AntMaze, 0.99 on other |
| CQL n actions | 10 |
| CQL $\alpha$ | 1 for Adroit, 5 for others |
| CQL max target backup | True |

Table 10: Common hyperparameters of TD3 and ReBRAC.

| Parameter | Value |
|---|---|
| optimizer | Adam |
| batch size | 1024 |
| learning rate | 1e-4 on AntMaze, 1e-3 on other |
| Q-function soft-update rate ($\tau$) | 5e-3 |
| discount factor ($\gamma$) | 0.999 on AntMaze, 0.99 on other |

Table 11: ReBRAC hyperparameters used in our experiments. All hyperparameters follow the best hyperparameters reported in Tarasov et al. (2023), except for the Kitchen domain, which was not included; for Kitchen, we performed a hyperparameter search and selected the best configuration.

| Task Name | $\beta_1$ (actor) | $\beta_2$ (critic) |
|---|---|---|
| halfcheetah-random | 0.001 | 0.1 |
| halfcheetah-medium | 0.001 | 0.01 |
| halfcheetah-medium-expert | 0.01 | 0.1 |
| halfcheetah-medium-replay | 0.01 | 0.001 |
| hopper-random | 0.001 | 0.01 |
| hopper-medium | 0.01 | 0.001 |
| hopper-medium-expert | 0.1 | 0.01 |
| hopper-medium-replay | 0.05 | 0.5 |
| walker2d-random | 0.01 | 0.0 |
| walker2d-medium | 0.05 | 0.1 |
| walker2d-medium-expert | 0.01 | 0.01 |
| walker2d-medium-replay | 0.05 | 0.01 |
| antmaze-large-play | 0.002 | 0.001 |
| antmaze-large-diverse | 0.002 | 0.002 |
| antmaze-ultra-diverse | 0.002 | 0.002 |
| door-binary | 0.1 | 0.01 |
| pen-binary | 0.1 | 0.01 |
| relocate-binary | 0.1 | 0.01 |
| kitchen-complete | 0.1 | 0.001 |
| kitchen-mixed | 0.1 | 0.001 |
| kitchen-partial | 0.1 | 0.001 |

Table 12: BC and FQE's hyperparameters.

| Parameter | Value |
|---|---|
| optimizer | Adam |
| batch size | 1024 |
| learning rate | 3e-4 |
| action | deterministic |
| FQE steps | 1e5 |
| Q-function soft-update rate ($\tau$) | 5e-3 |
| discount factor ($\gamma$) | 0.999 on AntMaze, 0.99 on other |

Table 13: Statistics (mean $\pm$ std) of offline datasets and pretrained policies. Pretrained policy scores $J(\pi_0)$ are reported as averages over 10 random seeds.

| Dataset | Dataset | | CalQL | | ReBRAC | | BC | |
|---|---|---|---|---|---|---|---|---|
| | $J(\pi_\mathcal{D})$ | #Trajs | $J(\pi_0)$ | order | $J(\pi_0)$ | order | $J(\pi_0)$ | order |
| halfcheetah-random-v2 | $-0.001 \pm 0.006$ | 1000 | $0.248 \pm 0.014$ | > | $0.275 \pm 0.011$ | > | $0.019 \pm 0.001$ | $\approx$ |
| halfcheetah-medium-replay-v2 | $0.271 \pm 0.135$ | 202 | $0.451 \pm 0.002$ | > | $0.504 \pm 0.003$ | > | $0.370 \pm 0.007$ | > |
| halfcheetah-medium-v2 | $0.406 \pm 0.029$ | 1000 | $0.470 \pm 0.003$ | > | $0.651 \pm 0.010$ | > | $0.427 \pm 0.002$ | $\approx$ |
| halfcheetah-medium-expert-v2 | $0.643 \pm 0.239$ | 2000 | $0.519 \pm 0.078$ | < | $1.010 \pm 0.019$ | > | $0.563 \pm 0.025$ | < |
| hopper-random-v2 | $0.012 \pm 0.005$ | 45240 | $0.091 \pm 0.017$ | > | $0.082 \pm 0.036$ | $\approx$ | $0.038 \pm 0.022$ | $\approx$ |
| hopper-medium-replay-v2 | $0.150 \pm 0.157$ | 2039 | $1.001 \pm 0.009$ | > | $0.965 \pm 0.042$ | > | $0.398 \pm 0.037$ | > |
| hopper-medium-v2 | $0.443 \pm 0.117$ | 2187 | $0.672 \pm 0.039$ | > | $1.016 \pm 0.012$ | > | $0.554 \pm 0.010$ | > |
| hopper-medium-expert-v2 | $0.648 \pm 0.319$ | 3214 | $1.058 \pm 0.108$ | > | $1.063 \pm 0.042$ | > | $0.556 \pm 0.012$ | < |
| walker2d-random-v2 | $0.000 \pm 0.001$ | 48908 | $0.082 \pm 0.043$ | > | $0.058 \pm 0.001$ | > | $0.008 \pm 0.001$ | $\approx$ |
| walker2d-medium-replay-v2 | $0.148 \pm 0.195$ | 1093 | $0.843 \pm 0.025$ | > | $0.853 \pm 0.045$ | > | $0.276 \pm 0.074$ | > |
| walker2d-medium-v2 | $0.620 \pm 0.239$ | 1191 | $0.742 \pm 0.071$ | > | $0.845 \pm 0.008$ | > | $0.507 \pm 0.073$ | < |
| walker2d-medium-expert-v2 | $0.826 \pm 0.285$ | 2191 | $1.073 \pm 0.035$ | > | $1.113 \pm 0.004$ | > | $1.075 \pm 0.004$ | > |
| pen-binary-v0 | $1.000 \pm 0.000$ | 846 | $0.657 \pm 0.059$ | < | $0.451 \pm 0.086$ | < | $0.589 \pm 0.068$ | < |
| door-binary-v0 | $1.000 \pm 0.000$ | 82 | $0.112 \pm 0.123$ | < | $0.000 \pm 0.000$ | < | $0.000 \pm 0.000$ | < |
| relocate-binary-v0 | $1.000 \pm 0.000$ | 36 | $0.010 \pm 0.011$ | < | $0.000 \pm 0.000$ | < | $0.000 \pm 0.000$ | < |
| kitchen-partial-v0 | $0.586 \pm 0.187$ | 600 | $0.764 \pm 0.094$ | > | $0.133 \pm 0.085$ | < | $0.222 \pm 0.079$ | < |
| kitchen-mixed-v0 | $0.598 \pm 0.145$ | 600 | $0.464 \pm 0.092$ | < | $0.034 \pm 0.033$ | < | $0.275 \pm 0.049$ | < |
| kitchen-complete-v0 | $1.000 \pm 0.000$ | 19 | $0.043 \pm 0.078$ | < | $0.002 \pm 0.004$ | < | $0.385 \pm 0.202$ | < |
| antmaze-large-diverse-v2 | $0.106 \pm 0.308$ | 999 | $0.305 \pm 0.055$ | > | $0.399 \pm 0.080$ | > | $0.000 \pm 0.000$ | < |
| antmaze-large-play-v2 | $0.105 \pm 0.307$ | 999 | $0.247 \pm 0.068$ | > | $0.351 \pm 0.072$ | > | $0.000 \pm 0.000$ | < |
| antmaze-ultra-diverse-v2 | $0.053 \pm 0.224$ | 999 | $0.118 \pm 0.072$ | $\approx$ | $0.127 \pm 0.132$ | $\approx$ | $0.000 \pm 0.000$ | $\approx$ |

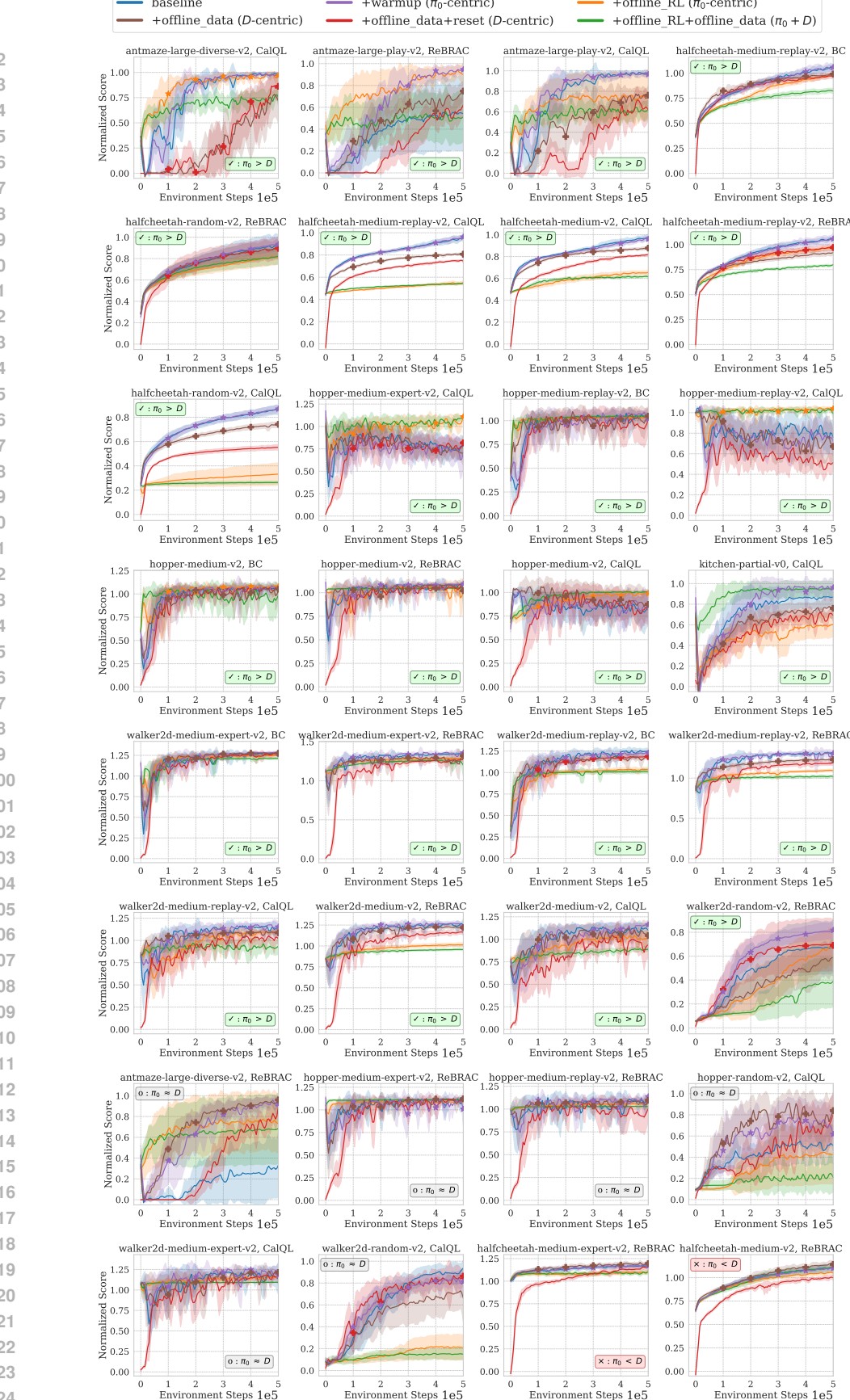

Figure 7: Full fine-tuning results in the **Superior** regime.

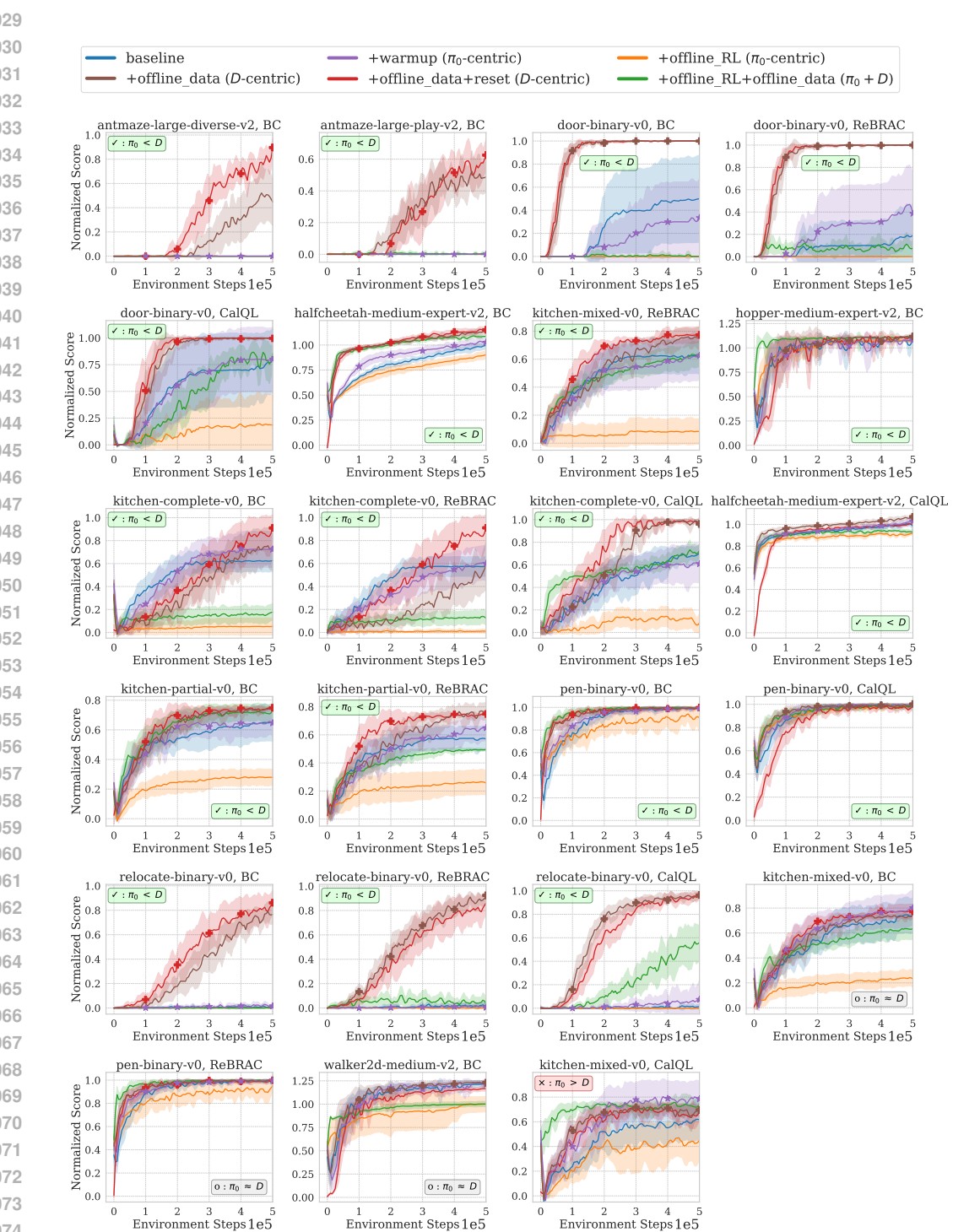

Figure 8: Full fine-tuning results in the **Inferior** regime.

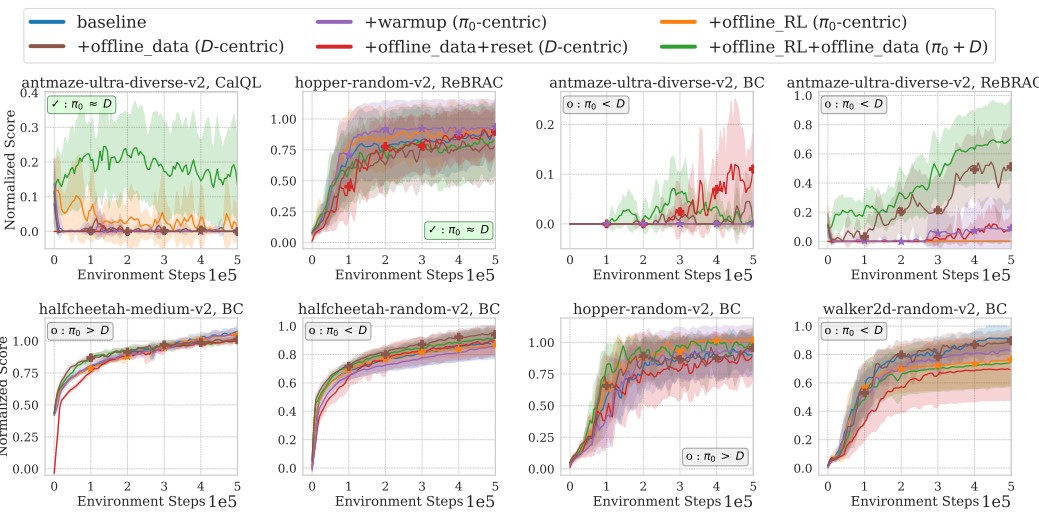

Figure 9: Full fine-tuning results in the **Comparable** regime.

# C    ALTERNATIVE TAXONOMIES OF THREE REGIMES

While our framework is defined using the returns of the pretrained agents and the dataset, it is also useful to consider alternative taxonomies based on different metrics. Below we present two such taxonomies, derived from Q-functions and behavior-cloning performance, respectively.

## C.1    Q-FUNCTION-BASED TAXONOMY

Conservative offline RL aims to outperform the behavior policy that generates the offline dataset (Levine et al., 2020). Ideally, the value function of the pretrained policy $\pi_0$ should therefore exceed that of the behavior policy $\pi_{\mathcal{D}}$ on in-dataset state-action pairs. Formally, Kostrikov et al. (2021, Lemma 2) show that for in-sample Q-learning, the optimal value function satisfies

$$Q^{\pi_0^*}(s, a) \geq Q^{\pi_{\mathcal{D}}}(s, a), \quad \forall (s, a) \in \mathcal{D}, \tag{7}$$

where $\pi_0^*$ denotes the optimal pretrained policy and $Q^\pi$ is the ground-truth value function of policy $\pi$. A weaker, expectation-based version of this condition is

$$\mathbb{E}_{(s,a)\sim\mathcal{D}}\Big[Q^{\pi_0^*}(s, a)\Big] \geq \mathbb{E}_{(s,a)\sim\mathcal{D}}[Q^{\pi_{\mathcal{D}}}(s, a)]. \tag{8}$$

Motivated by this result, we examine a taxonomy based on comparing $\mathbb{E}_{(s,a)\sim\mathcal{D}}[\hat{Q}^{\pi_0}(s, a)]$ and $\mathbb{E}_{(s,a)\sim\mathcal{D}}[Q^{\pi_{\mathcal{D}}}(s, a)]$, where $\hat{Q}^{\pi_0}$ denotes the pretrained Q-function. The intuition is that if

$$\mathbb{E}_{(s,a)\sim\mathcal{D}}[\hat{Q}^{\pi_0}(s, a)] \geq \mathbb{E}_{(s,a)\sim\mathcal{D}}[Q^{\pi_{\mathcal{D}}}(s, a)],$$

then pretraining approximately satisfies Eq. 8, placing it in the **Superior** or **Comparable** regime; otherwise, it falls into the **Inferior** regime.

In practice, we estimate $Q^{\pi_{\mathcal{D}}}(s, a)$ using the Monte Carlo return $G(s, a)$ from $\mathcal{D}$. We then perform a $t$-test to assess whether $\mathbb{E}_{(s,a)\sim\mathcal{D}}\Big[\hat{Q}^{\pi_0}(s, a) - G(s, a)\Big]$ is zero. Acceptance of the null corresponds to the **Comparable** Q-regime; a significantly positive difference indicates the **Superior** Q-regime, and a significantly negative difference indicates the **Inferior** Q-regime.

Finally, we evaluate how well this Q-based taxonomy aligns with the fine-tuning results. The corresponding confusion matrix is reported in Table 14. It achieves 32 out of 63 correct predictions (**51%**), 19 opposite mismatches (30%), and 12 adjacent mismatches (19%). Although this accuracy is substantially better than random guessing (33%), it remains noticeably lower than the performance of our framework (**71%**).

Table 14: Confusion matrix of fine-tuning results using **Q-based taxonomy**. Green cells: correct predictions (32/63); red cells: opposite mismatches (19/63); gray cells: adjacent mismatches (12/63). Overall, Q-based taxonomy achieve 51% correct predictions with 30% opposite mismatches.

|  |  | Pretraining Q-based Regime | | |
|---|---|---|---|---|
|  |  | **Superior** | **Comparable** | **Inferior** |
| **Fine-tune** | $\pi_0$-centric $>$ $\mathcal{D}$-centric | **23** | 0 | 4 |
|  | $\pi_0$-centric $\approx$ $\mathcal{D}$-centric | 8 | **0** | 3 |
|  | $\pi_0$-centric $<$ $\mathcal{D}$-centric | 15 | 1 | **9** |

## C.2    BEHAVIOR-CLONING-BASED TAXONOMY

Instead of relying on the return of the behavior policy $J(\pi_{\mathcal{D}})$, one can define a taxonomy based on the performance of a behavior-cloned policy $\pi_{\text{BC}}$ learned on the offline dataset $\mathcal{D}$.

Similar to our taxonomy, we use a $t$-test with a margin of $\delta = 0.05$ to assess whether $J(\pi_{\text{BC}})$ and $J(\pi_{\mathcal{D}})$ differ significantly, thereby identifying the three corresponding regimes. The corresponding

confusion matrix is shown in Table 15. The BC-based taxonomy achieves 26 out of 63 correct predictions (**41%**), 3 opposite mismatches (5%), and 34 adjacent mismatches (54%). While better than random guessing, this result remains noticeably lower than the performance of our framework (**71%**).

Table 15: Confusion matrix of fine-tuning results using **BC-based taxonomy**. Green cells: correct predictions (26/63); red cells: opposite mismatches (34/63); gray cells: adjacent mismatches (3/63). Overall, BC-based taxonomy achieve 41% correct predictions with 5% opposite mismatches.

| | | BC-based Regime | | |
|---|---|---|---|---|
| | | Superior | Comparable | Inferior |
| **Fine-tune** | $\pi_0$-centric $>$ $\mathcal{D}$-centric | **17** | 10 | 0 |
| | $\pi_0$-centric $\approx$ $\mathcal{D}$-centric | 4 | **6** | 1 |
| | $\pi_0$-centric $<$ $\mathcal{D}$-centric | 3 | 19 | **3** |

