# OpenReview forum: "The Three Regimes of Offline-to-Online Reinforcement Learning"
_ICLR.cc/2026/Conference — Submitted to ICLR 2026_

### Official Review · Reviewer_FHZW · 2025-10-18

**Soundness:** 2
**Presentation:** 3
**Contribution:** 2
**Rating:** 4
**Confidence:** 4

**Summary:**

The paper conducts an empirical study over 21 D4RL tasks to analyze how the performance of offline pre-trained policies influence the performances of fine-tuning algorithms. In particular, the authors classify the fine-tuning algorithms based on how much it relies on the pre-trained policy vs. offline dataset (i.e., $\pi_0$-centric vs. $\mathcal{D}$-centric) and uncover a trend where $\pi$-centric fine-tuning algorithms tend to work well when pre-trained policy has better performance compared to that of the behavior policy, and vice versa (i.e., $\mathcal{D}$-centric fine-tuning algorithms tend to work well when the pre-trained policy has poor performance compared to that of the behavior policy). The authors also define some notions of prior knowledge, stability and plasticity to explain the trend. In particular, the authors propose the stability-plasticity principle which suggests that (1) for tasks where the pre-trained policy has poor performance, the fine-tuning algorithms need to have sufficient plasticity, and (2) for tasks where the pre-trained policy has good performance, the fine-tuning algorithms need to have sufficient stability. $\mathcal{D}$-centric methods and $\pi_0$-centric methods prioritize plasticity and stability respectively, explaning the empirical synergies with tasks with different pre-training performance.

**Strengths:**

- The empirical study is thorough and provides convincing evidence that the performance of the pre-training policy can be used as a relatively robust predictor on (43 of 63 cases on D4RL tasks) which class of algorithms ($\pi_0$-centric vs. $\mathcal{D}$-centric) would work the best for online fine-tuning.


- The selected $\pi_0$-centric and $\mathcal{D}$-centric algorithms are strong algorithms in the literature, making the empirical study fair and reasonable, which is valuable to the community.

**Weaknesses:**

My main concerns about the paper are two-fold.


*(1) Regime classification for the tasks (Superior vs. Comparable vs. Inferior) is conceptually flawed and may be subtly incorrect in many scenarios.*


The authors propose to classify the tasks based on the return of the pre-trained policy $J(\pi_0)$ and the return of the behavior policy $J(\pi_\mathcal{D})$ and then argue that poor performing policy needs data centric algorithms with high plasticity (and often trades off stability). While this may work well for many cases empirically, it is based on a key assumption that the performance of the policy is indicative of its usefulness in generating data for online fine-tuning. For example, this assumption can break down in many long-horizon, sparse reward tasks where it is difficult to pre-train a policy to achieve non-trivial success rate directly, but the policy can progress in the task far enough to provide good exploratory data. This is where stability should be preferred over plasticity, but can be overlooked if only the returns are being examined.


While it is true that finding a very principled classification can be quite challenging, relying the classification entirely on the expected returns is a bit too simplistic/crude of an approximation that is expected to not be very future proof (e.g., especially as we begin to study more challenging and complex tasks).


*(2) The authors attempt to formalize the principle of plasticity and stability, the main intuition and explanation behind the performance predictions of online fine-tuning algorithms, with mathematical definitions, but these quantities are not being evaluated in the empirical study.*


The paper makes three formal definitions: (1) prior knowledge, (2) stability, and (3) plasticity: Prior knowledge is the larger of the pre-trained policy’s return and the behavior policy’s return; stability and plasticity measure the how much the performance of the policy fluctuate over the course of training. In particular, the stability metric takes a minimum over the returns of $N$ policies over the course of online fine-tuning (at an unknown interval). This metric conceptually can be very sensitive to the choice of $N$ because it can be seen as taking the minimum of $N$ random variables (same for plasticity), which can make it hard to quantify to what degree the online fine-tuning algorithms exhibit stability vs. plasticity. Furthermore, I could not find any empirical value of these metrics from the paper to justify these definitions.


Overall, without a more rigorous discussion of these definitions or empirical evidence that they correspond to the intuitions, these definitions in their current form add little value to the paper.

**Questions:**

1. What are the values of prior knowledge, stability and plasticity for each class of the algorithms (e.g., $\pi_0$-centric, data centric)?

2. “We fixed a limited set of representative variants and hyperparameters across all settings” – how are the hyperparameters chosen? For these analysis results, it is very important to perform proper tuning of all the algorithms in the study to mitigate hyperparameter biases. Otherwise, the results may be explainable by just an artifact of a biased hyperparameter selection.

---

> ### Author Response · Authors · 2025-11-25
> **Response to FHZW-1**
>
> We are grateful for your comprehensive review and valuable feedback. We will address each of your comments and concerns in the following responses, as well as in our revised manuscript.
>
> ---
>
> > **W1**: Regime classification for the tasks (Superior vs. Comparable vs. Inferior) is conceptually flawed and may be subtly incorrect in many scenarios. ... While this may work well for many cases empirically, it is based on a key assumption that the performance of the policy is indicative of its usefulness in generating data for online fine-tuning. For example, this assumption can break down in many long-horizon, sparse reward tasks where it is difficult to pre-train a policy to achieve non-trivial success rate directly, but the policy can progress in the task far enough to provide good exploratory data. This is where stability should be preferred over plasticity, but can be overlooked if only the returns are being examined.
>
> Summary: (1) We present empirical results on long-horizon, sparse-reward tasks; (2) We offer justification for our return-based formulation and acknowledge that alternative proxy metrics, when accessible, could provide additional benefits.
>
>
> **(1) Empirical evidence on long-horizon, sparse-reward tasks.** In our experiments, we conducted evaluations on *long-horizon, sparse-reward* settings, including 3 AntMaze tasks, 3 Adroit manipulation tasks, and 3 Kitchen tasks.
> - AntMaze: long-horizon navigation tasks that require controlling an 8-DoF Ant robot to reach a goal.
> - Adroit: dexterous manipulation tasks involving a 28-DoF five-fingered robotic hand that must manipulate a pen to a target location, open a door, or relocate a ball to a desired position.
> - Kitchen: long-horizon manipulation tasks in which a 9-DoF Franka robot arm must perform *four* sequential subtasks in a simulated kitchen, and the reward at each step is the number of subtasks completed.
>
> The confusion matrices for these domains are presented below. Our regime classification achieves **77.8\% (21/27)** correct predictions with only **3.7\% (1/27)** opposite mismatch across the three long-horizon sparse-reward domains.
>
> **Table: Confusion matrix for all sparse-reward, long-horizon tasks (AntMaze, Adroit, Kitchen)**
> |fine-tuning \ regime | Superior | Comparable | Inferior |
> |-|-|-|-|
> | $\pi_0$-centric $>$ $\mathcal{D}$-centric | **4**    | 1| 0|
> | $\pi_0$-centric $\approx$ $\mathcal{D}$-centric | 0| **1**| 2|
> | $\pi_0$-centric $<$ $\mathcal{D}$-centric | 1        | 2| **16**|
>
> **Table (breakdown): AntMaze**
> |fine-tuning \ regime | Superior | Comparable | Inferior |
> |-|-|-|-|
> | $\pi_0$-centric $>$ $\mathcal{D}$-centric| **3**    | 1| 0|
> | $\pi_0$-centric $\approx$ $\mathcal{D}$-centric| 0| **1**| 2|
> | $\pi_0$-centric $<$ $\mathcal{D}$-centric| 0| 0| **2**    |
>
> **Table (breakdown): Adroit**
> | fine-tuning \ regime   | Superior | Comparable | Inferior |
> |-|-|-|-|
> | $\pi_0$-centric $>$ $\mathcal{D}$-centric| 0| 0| 0|
> | $\pi_0$-centric $\approx$ $\mathcal{D}$-centric    | 0| 0| 0|
> | $\pi_0$-centric $<$ $\mathcal{D}$-centric| 0| 1| **8**    |
>
> **Table (breakdown): Kitchen**
> | fine-tuning \ regime | Superior | Comparable | Inferior |
> |-|-|-|-|
> | $\pi_0$-centric $>$ $\mathcal{D}$-centric| **1**    | 0| 0|
> | $\pi_0$-centric $\approx$ $\mathcal{D}$-centric    | 0| 0| 0|
> | $\pi_0$-centric $<$ $\mathcal{D}$-centric| 1| 1| **6**    |
>
> (2)  Furthermore, we use the average return as a simple yet effective metric to quantify the usefulness of the pretrained policy $\pi_0$ and offline dataset $\mathcal D$. It is true that in some situations, such as long-horizon sparse-reward settings, the return alone may not fully reflect the usefulness of a policy in generating data for online fine-tuning. Long-horizon sparse rewards also make it inherently difficult for RL to learn a good policy. If additional proxy metrics are available (e.g. dense reward), we agree that incorporating them could make our framework more robust and accurate. However, in the current setting, we **do not assume access to such metrics**. We acknowledge this limitation of our framework and have revised the manuscript accordingly.

---

> ### Author Response · Authors · 2025-11-25
> **Response to FHZW-2**
>
> >**W2-1**: ...In particular, the stability metric takes a minimum over the returns of $N$ policies over the course of online fine-tuning (at an unknown interval). This metric conceptually can be very sensitive to the choice of $N$ because it can be seen as taking the minimum of $N$ random variables (same for plasticity), which can make it hard to quantify to what degree the online fine-tuning algorithms exhibit stability vs. plasticity.
>
> The metrics were defined under an idealized assumption in which $N$ denotes the number of gradient steps during fine-tuning, as each gradient update results in a new policy. However, in practice it is not feasible to evaluate the agent at such a high frequency. In our emprical studies, we evaluated the performance of agent every 5000 environment steps during online fine-tuning, ensuring a fair comparison across all fine-tuning methods.
>
> ---
>
> > **W2-2**: ... but these quantities are not being evaluated in the empirical study.
> >**Q1**: What are the values of prior knowledge, stability and plasticity for each class of the algorithms?
>
> We report the empirical values of stability, plasticity, and the resulting improvement over $J^*_{off}$ (given by stability plus plasticity) in our empirical studies, as presented below.
>
> **We observe that the empirical values of stability and plasticity align well with intuition**:
>
> 1. **Stability**: the “offline RL + offline data” method achieves the *strongest stability* across all regimes, while the baseline yields the *lowest stability* (except for “offline data + reset,” which performs a full parameter reset).
> 2. **The strongest plasticity is *regime-dependent***: In terms of plasticity, we find that $\mathcal{D}$-centric methods provide the strongest plasticity in the Inferior regime, since the offline data contain high-quality trajectories that are useful for online fine-tuning, but these methods show lower plasticity in the Superior regime. A similar pattern appears for the “warmup” setting: it offers strong plasticity in the Superior regime but lower plasticity in the Inferior regime.
> 3. **The effect of fine-tuning methods are *regime-specific***. Aggregated results show modest differences across methods, but regime-specific tables reveal clear, method-dependent patterns, highlighting the practical utility of the stability–plasticity principle and the three-regime framework.

---

> ### Author Response · Authors · 2025-11-25
> **Response to FHZW-2 (table)**
>
> **Table: Empirical values of Stability and Plasticity in Inferior Regime**
>
> Highest values are bolded (for plasticity, the second-highest is bolded since the reset method is always the highest).
> | Fine-tuning method| Stability| Plasticity| Improvement = Stability + Plasticity |
> |--|--|--|--|
> | baseline| -0.711 ± 0.305| 0.561 ± 0.378| -0.150 ± 0.467|
> | + warmup ($\pi_0$-centric) | -0.692 ± 0.302| 0.577 ± 0.382| -0.116 ± 0.475                         |
> | + offline RL ($\pi_0$-centric)| -0.659 ± 0.319 | 0.297 ± 0.292| -0.362 ± 0.498|
> | + offline data ($\mathcal{D}$-centric)| -0.694 ± 0.304 | **0.849 ± 0.188**| 0.155 ± 0.278|
> | + offline data + reset ($\mathcal{D}$-centric)| -0.769 ± 0.275 | 0.951 ± 0.138| **0.182 ± 0.275**|
> | + offline RL + offline data (mixed $\pi_0+\mathcal{D}$)| **-0.616 ± 0.352** | 0.475 ± 0.284 | -0.142 ± 0.437 |
>
>
> **Table: Empirical values of Stability and Plasticity in Superior Regime**
> | Fine-tuning method| Stability| Plasticity| Improvement = Stability + Plasticity |
> |--|---|--|--|
> | baseline| -0.296 ± 0.329| 0.834 ± 0.275| 0.538 ± 0.271 |
> | + warmup ($\pi_0$-centric)| -0.296 ± 0.336| **0.883 ± 0.242**|**0.587 ± 0.267** |
> | + offline RL  ($\pi_0$-centric)| -0.158 ± 0.250| 0.521 ± 0.292| 0.363 ± 0.261 |
> | + offline data ($\mathcal{D}$-centric)| -0.292 ± 0.346     | 0.820 ± 0.270| 0.528 ± 0.258 |
> | + offline data + reset ($\mathcal{D}$-centric)| -0.467 ± 0.325| 1.026 ± 0.175| 0.559 ± 0.290 |
> | + offline RL + offline data (mixed $\pi_0+\mathcal{D}$)| **-0.104 ± 0.211** | 0.394 ± 0.288| 0.290 ± 0.242 |
>
>
> **Table: Empirical values of Stability and Plasticity in Comparable Regime**
> | Fine-tuning method| Stability| Plasticity| Improvement = Stability + Plasticity |
> |-|--|---|--|
> | baseline| -0.030 ± 0.047| 0.573 ± 0.420| 0.543 ± 0.452 |
> | + warmup ($\pi_0$-centric)| -0.029 ± 0.047| 0.562 ± 0.401| 0.533 ± 0.432 |
> | + offline RL ($\pi_0$-centric)| -0.030 ± 0.047| 0.575 ± 0.387| 0.545 ± 0.414 |
> | + offline data  ($\mathcal{D}$-centric)| -0.029 ± 0.047| 0.694 ± 0.347| 0.664 ± 0.375 |
> | + offline data + reset ($\mathcal{D}$-centric)| -0.087 ± 0.136| 0.674 ± 0.380| 0.587 ± 0.386 |
> | + offline RL + offline data (mixed $\pi_0+\mathcal{D}$)| -0.024 ± 0.037 | 0.692 ± 0.307      | 0.668 ± 0.326 |
>
> **Table: Empirical values of Stability and Plasticity in all regimes**
> | Fine-tuning method| Stability| Plasticity| Improvement = Stability + Plasticity |
> |-|--|---|--|
> | baseline| -0.414 ± 0.384| 0.701 ± 0.361| 0.287 ± 0.503|
> | + warmup ($\pi_0$-centric)| -0.407 ± 0.380| 0.730 ± 0.356| 0.323 ± 0.503|
> | + offline RL ($\pi_0$-centric)| -0.324 ± 0.367         | 0.446 ± 0.326| 0.122 ± 0.533|
> | + offline data  ($\mathcal{D}$-centric)| -0.405 ± 0.387| 0.814 ± 0.259| 0.409 ± 0.344|
> | + offline data + reset  ($\mathcal{D}$-centric)| -0.529 ± 0.362| 0.954 ± 0.230| 0.425 ± 0.350|
> | + offline RL + offline data  (mixed $\pi_0+\mathcal{D}$)| -0.281 ± 0.365| 0.461 ± 0.304| 0.181 ± 0.432|

---

> ### Author Response · Authors · 2025-11-25
> **Response to FHZW-3**
>
> > **Q2**: how are the hyperparameters chosen?
>
> - For both SAC and CalQL, our experiments follow the implementations and hyperparameters provided in the [WSRL codebase](https://github.com/zhouzypaul/wsrl).
> - For TD3, we follow the hyperparameters reported in the TD3 paper and used in the [official implementation](https://github.com/sfujim/TD3). For ReBRAC, we adopt the hyperparameters reported in the ReBRAC paper and implemented in the [official codebase](https://github.com/tinkoff-ai/ReBRAC). As ReBRAC uses environment-specific regularization coefficients, for tasks where the paper did not report these values (e.g., all Kitchen tasks), we conducted a hyperparameter search over {0.1, 0.01, 0.002, 0.001} for both the actor and critic coefficients and selected the best-performing combination for each task.
> - For BC, we implement a deterministic BC policy using the same network architecture as the TD3 and ReBRAC agents. Once the BC actor is trained, we perform Fitted Q Evaluation (FQE) [1]  on the offline dataset to learn a corresponding critic. These actor and critic networks then serve as the initialization for the subsequent online RL fine-tuning phase.
>
> We also provide the hyperparameters used in our experiments in the appendix B.3 of the revised manuscript.
>
> Reference:
>
> [1] Empirical study of off-policy policy evaluation for reinforcement learning. arXiv preprint arXiv:1911.06854
>
> ---
>
> We hope that our response can address the reviewer’s concerns, and we are happy to provide further details should the reviewer find any aspect insufficiently explained.

---

> > ### Comment · Reviewer_FHZW · 2025-11-26
> >
> > Thanks for providing additional information on hyperparameters and additional results on the stability/plasticity metrics. I have a few follow-up questions.
> >
> > ## 1. Why is reset not bolded for plasticity?
> >
> > It seems a bit odd to me that the reset method is treated as a special case because this is potentially a case where it breaks the intuition that the paper tries to argue. The strongest plasticity is achieved by reset *regardless* of which regime it is under.
> >
> > ## 2. Stability-plasticity results do not align with the proposed explanation.
> >
> > The paper mentions that "In the Superior Regime, stability relative to $\pi_0$ should be prioritized, meaning that online updates should avoid degrading below the performance already achieved by the pretrained policy". However, in the table above, the method that achieves the highest stability (e.g., "+ offline RL ($\pi_0$-centric)", "+ offline RL + offline data (mixed $\pi_0$ + $\mathcal{D}$)") actually perform the worst. This casts doubts on whether the intuitive explanations provided actually align with what is going on in practice. Furthermore, plasticity seems to be correlate better with the performance (e.g., in both Superior and Inferior regime as shown in the tables above).
> >
> > Overall, I think the stability-plasticity decomposition/framework offers an interesting explanations of the empirical results for $\pi_0$ and $\mathcal{D}$-centric methods, but the current results do not seem to align with the explanations and it is unclear to me how they can be used in practice.

---

> > > ### Author Response · Authors · 2025-12-03
> > > **Response to follow-up questions-1**
> > >
> > > Thank you for your follow-up questions.
> > >
> > > First, we identified a minor mistake in the aggregation of the empirical values. We have corrected it and present the corrected results in the following tables. Importantly, this fix does *not* affect our conclusions. Since the number of fine-tuning steps can influence the empirical values, we report results at both an **early stage (50k steps)** and a **late stage (500k steps)**, attached at the end.
> > >
> > > > Q1: Why is reset not bolded for plasticity? It seems a bit odd to me that the reset method is treated as a special case because this is potentially a case where it breaks the intuition that the paper tries to argue. The strongest plasticity is achieved by reset regardless of which regime it is under.
> > >
> > > **(1) Clarification on the reset method.** We have now bolded “reset” for plasticity. Previously, we did not highlight it because it systematically achieves the highest plasticity in almost all settings and we focused on contrasting methods whose behavior is more informative. Importantly, the “reset” result is **consistent with our intuition**: a full parameter reset makes $\min_j J(\pi_j) \approx 0$, maximizing plasticity, but comes at the cost of the lowest possible stability.
> > >
> > > **(2) Refined observations.** We have updated the text to clearly identify the best-performing method along each metric:
> > >
> > > - **Stability and Plasticity**: The "offline RL + offline data" method attains the highest average stability across regimes, as it improves stability relative to both $\pi_0$ and $\mathcal{D}$. It also achieves the highest plasticity in the Comparable regime. In contrast, "offline data + reset" yields the highest average plasticity in the Superior and Inferior regimes, but this comes at the cost of driving stability to its lowest possible level due to the full parameter reset.
> > > - **Improvement**: The highest average improvement is regime-dependent. In the Superior regime, it is achieved by "warmup", which is a $\pi_0$-centric method; in the Inferior regime, it is achieved by "offline data + reset", which is a $\mathcal{D}$-centric method. In the Comparable regime, the best-performing method is "offline RL + offline data", with a small advantage. These patterns are consistent with intuition: in the Superior regime, $\pi_0$ provides more valuable knowledge to leverage, whereas in the Inferior regime, the offline dataset $\mathcal{D}$ contains the more valuable knowledge.
> > >
> > >
> > >
> > > ---
> > >
> > > > Q2-1: The paper mentions that "In the Superior Regime, stability relative to $\pi_0$ should be prioritized, meaning that online updates should avoid degrading below the performance already achieved by the pretrained policy". However, the method that achieves the highest stability actually perform the worst in the Superior regime."
> > >
> > > **(1)** We agree that the original wording on "avoid degrading" was unclear. We have revised it to:
> > >
> > > "In the Superior regime, stability relative to $\pi_0$ should be *prioritized* over stability relative to $\mathcal{D}$, because $\pi_0$ contains more useful knowledge than $\mathcal{D}$ ... At the same time, maintaining sufficient plasticity across all regimes is essential for efficient fine-tuning."
> > >
> > >
> > > **(2)** It is unsurprising that the most stable method performs the worst, given the fundamental trade-off between stability and plasticity. The "offline RL + offline data" method enforces very high stability by improving stability with respect to both the offline dataset and the pretrained policy, but this suppresses plasticity and therefore limits the improvement during fine-tuning.
> > >
> > > An extreme example can also illustrate this: if we maximize stability by *not fine-tuning at all*, the policy remains $\pi_0$. This configuration achieves the highest possible stability but also yields the **lowest possible improvement**. Since improvement can be expressed as $\max_i J(\pi_i)$-$J_{off}$* , so the minimum achievable improvement is $J(\pi_0) - J^*_{off}$.

---

> > > > ### Author Response · Authors · 2025-12-03
> > > > **Response to follow-up questions-2**
> > > >
> > > > > Q2-2: Furthermore, plasticity seems to be correlate better with the performance (e.g., in both Superior and Inferior regime as shown in the tables above).
> > > >
> > > > We agree that “plasticity seems to correlate better with performance,” and this is expected because plasticity measures the potential performance gain relative to the worst policy encountered.
> > > >
> > > > However, **this pattern does not hold universally** and we provide two concrete counterexamples: (1) superior regime, (2) the Inferior regime at the early stage.
> > > >
> > > > **(1)** Superior regime.
> > > > The "offline data + reset" method attains the highest plasticity, yet yields the *lowest* improvement at 50k steps and only moderate improvement at 500k steps. This shows that when $\pi_0$ is strong, plasticity alone does not determine performance; maintaining stability is essential.
> > > >
> > > > **(2)** Inferior regime, early stage (50k steps).
> > > > At an early stage of fine-tuning, the method with the highest average improvement in Inferior regime is "offline RL + offline data", which has the *highest* stability and the *third-highest* plasticity. This again shows that plasticity does not strictly dominate improvement across all settings.
> > > >
> > > > These cases illustrate that while plasticity often correlates with improvement, it is not the sole determinant—better improvement depends on the balance between stability and plasticity. Moreover, the relative importance of stability and plasticity varies across regimes and stages of fine-tuning.
> > > >
> > > > > The current results do not seem to align with the explanations and it is unclear to me how they can be used in practice.
> > > >
> > > > We believe the clarifications and new analyses above directly address this concern. After correcting a minor aggregation issue and providing both early-stage and late-stage analyses, the empirical trends now align cleanly with the explanations: stability and plasticity exhibit the expected trade-off, plasticity often correlates with improvement but does not universally dominate, and the best improvement depends on both the regime and the stage of fine-tuning.
> > > >
> > > > Regarding practical use, our framework offers actionable guidance for offline-to-online RL. By identifying two sources of stability—$\pi_0$ and $\mathcal{D}$—and show that $\pi_0$-centric methods are preferred in the Superior regime, while $\mathcal{D}$-centric methods are favored in the Inferior regime. This offers practitioners **clearer guidance on selecting appropriate design choices and methods for each setting based on its regime**, rather than applying a single approach uniformly across all tasks.

---

> ### Author Response · Authors · 2025-12-03
> **Response to follow-up questions-Corrected Table-1**
>
> ## Corrected Table: Empirical values of Stability and Plasticity in Superior Regime
> ### 50K steps
> | Fine-tuning method | Stability | Plasticity | Improvement |
> |-|-|-|-|
> | baseline | $-0.352 \pm 0.344$ | $0.525 \pm 0.325$ | $0.172 \pm 0.145$ |
> | + warmup ($\pi_0$-centric) | $-0.328 \pm 0.348$ | $0.501 \pm 0.315$ | $\mathbf{0.173 \pm 0.122}$ |
> | + offline RL ($\pi_0$-centric) | $-0.162 \pm 0.242$ | $0.289 \pm 0.255$ | $0.127 \pm 0.151$ |
> | + offline data ($\mathcal{D}$-centric) | $-0.293 \pm 0.351$ | $0.448 \pm 0.318$ | $0.155 \pm 0.138$ |
> | + offline data + reset ($\mathcal{D}$-centric) | $-0.615 \pm 0.338$ | $\mathbf{0.581 \pm 0.349}$ | $-0.034 \pm 0.311$ |
> | + offline RL + offline data (mixed $\pi_0+\mathcal{D}$) | $\mathbf{-0.072 \pm 0.128}$ | $0.204 \pm 0.189$ | $0.132 \pm 0.138$ |
>
> ### 500K steps
> | Fine-tuning method | Stability | Plasticity | Improvement |
> |-|-|-|-|
> | baseline | $-0.399 \pm 0.363$ | $0.832 \pm 0.275$ | $0.433 \pm 0.276$ |
> | + warmup ($\pi_0$-centric) | $-0.394 \pm 0.371$ | $0.865 \pm 0.251$ | $\mathbf{0.471 \pm 0.258}$ |
> | + offline RL ($\pi_0$-centric) | $-0.199 \pm 0.268$ | $0.519 \pm 0.291$ | $0.319 \pm 0.256$ |
> | + offline data ($\mathcal{D}$-centric) | $-0.376 \pm 0.380$ | $0.796 \pm 0.282$ | $0.421 \pm 0.251$ |
> | + offline data + reset ($\mathcal{D}$-centric) | $-0.615 \pm 0.338$ | $\mathbf{1.011 \pm 0.183}$ | $0.396 \pm 0.271$ |
> | + offline RL + offline data (mixed $\pi_0+\mathcal{D}$) | $\mathbf{-0.124 \pm 0.213}$ | $0.393 \pm 0.288$ | $0.270 \pm 0.233$ |
>
> ---
>
> ## Corrected Table: Empirical values of Stability and Plasticity in Inferior Regime
> ### 50K steps
> | Fine-tuning method | Stability | Plasticity | Improvement |
> |-|-|-|-|
> | baseline | $-0.688 \pm 0.348$ | $0.216 \pm 0.219$ | $-0.472 \pm 0.408$ |
> | + warmup ($\pi_0$-centric) | $-0.671 \pm 0.343$ | $0.213 \pm 0.217$ | $-0.458 \pm 0.422$ |
> | + offline RL ($\pi_0$-centric) | $-0.665 \pm 0.342$ | $0.159 \pm 0.197$ | $-0.505 \pm 0.428$ |
> | + offline data ($\mathcal{D}$-centric) | $-0.670 \pm 0.343$ | $0.268 \pm 0.215$ | $-0.402 \pm 0.398$ |
> | + offline data + reset ($\mathcal{D}$-centric) | $-0.832 \pm 0.191$ | $\mathbf{0.417 \pm 0.301}$ | $-0.415 \pm 0.395$ |
> | + offline RL + offline data (mixed $\pi_0+\mathcal{D}$) | $\mathbf{-0.640 \pm 0.357}$ | $0.266 \pm 0.219$ | $\mathbf{-0.374 \pm 0.415}$ |
>
> ### 500K steps
> | Fine-tuning method | Stability | Plasticity | Improvement |
> |-|-|-|-|
> | baseline | $-0.692 \pm 0.307$ | $0.475 \pm 0.378$ | $-0.217 \pm 0.480$ |
> | + warmup ($\pi_0$-centric) | $-0.677 \pm 0.306$ | $0.495 \pm 0.389$ | $-0.182 \pm 0.490$ |
> | + offline RL ($\pi_0$-centric) | $-0.650 \pm 0.322$ | $0.262 \pm 0.283$ | $-0.388 \pm 0.489$ |
> | + offline data ($\mathcal{D}$-centric) | $-0.674 \pm 0.308$ | $0.696 \pm 0.340$ | $0.023 \pm 0.451$ |
> | + offline data + reset ($\mathcal{D}$-centric) | $-0.769 \pm 0.275$ | $\mathbf{0.870 \pm 0.299}$ | $\mathbf{0.101 \pm 0.205}$ |
> | + offline RL + offline data (mixed $\pi_0+\mathcal{D}$) | $\mathbf{-0.614 \pm 0.354}$ | $0.440 \pm 0.297$ | $-0.174 \pm 0.441$ |
>
> ---
>
> ## Corrected Table: Empirical values of Stability and Plasticity in Comparable Regime
> ### 50K steps
> | Fine-tuning method | Stability | Plasticity | Improvement |
> |-|-|-|-|
> | baseline | $-0.083 \pm 0.084$ | $0.163 \pm 0.237$ | $0.080 \pm 0.251$ |
> | + warmup ($\pi_0$-centric) | $-0.082 \pm 0.085$ | $0.163 \pm 0.195$ | $0.081 \pm 0.212$ |
> | + offline RL ($\pi_0$-centric) | $-0.079 \pm 0.085$ | $0.214 \pm 0.264$ | $0.135 \pm 0.270$ |
> | + offline data ($\mathcal{D}$-centric) | $-0.081 \pm 0.085$ | $0.147 \pm 0.161$ | $0.066 \pm 0.166$ |
> | + offline data + reset ($\mathcal{D}$-centric) | $-0.114 \pm 0.147$ | $\mathbf{0.261 \pm 0.250}$ | $0.147 \pm 0.237$ |
> | + offline RL + offline data (mixed $\pi_0+\mathcal{D}$) | $\mathbf{-0.057 \pm 0.058}$ | $0.240 \pm 0.251$ | $\mathbf{0.182 \pm 0.258}$ |
>
> ### 500K steps
> | Fine-tuning method | Stability | Plasticity | Improvement |
> |-|-|-|-|
> | baseline | $-0.043 \pm 0.071$ | $0.560 \pm 0.411$ | $0.516 \pm 0.452$ |
> | + warmup ($\pi_0$-centric) | $-0.042 \pm 0.072$ | $0.543 \pm 0.390$ | $0.501 \pm 0.423$ |
> | + offline RL ($\pi_0$-centric) | $-0.043 \pm 0.072$ | $0.564 \pm 0.382$ | $0.521 \pm 0.416$ |
> | + offline data ($\mathcal{D}$-centric) | $-0.042 \pm 0.072$ | $0.680 \pm 0.338$ | $0.638 \pm 0.361$ |
> | + offline data + reset ($\mathcal{D}$-centric) | $-0.106 \pm 0.139$ | $0.623 \pm 0.431$ | $0.516 \pm 0.431$ |
> | + offline RL + offline data (mixed $\pi_0+\mathcal{D}$) | $\mathbf{-0.035 \pm 0.061}$ | $\mathbf{0.682 \pm 0.302}$ | $\mathbf{0.646 \pm 0.314}$ |

---

> > ### Author Response · Authors · 2025-12-03
> > **Response to follow-up questions-Corrected Table-2**
> >
> > ## Corrected Table: Empirical values of Stability and Plasticity in all regimes
> >
> > ### 50K steps
> > | Fine-tuning method | Stability | Plasticity | Improvement |
> > |-|-|-|-|
> > | baseline | $-0.433 \pm 0.379$ | $0.397 \pm 0.329$ | $-0.036 \pm 0.395$ |
> > | + warmup ($\pi_0$-centric) | $-0.413 \pm 0.380$ | $0.382 \pm 0.315$ | $-0.031 \pm 0.389$ |
> > | + offline RL ($\pi_0$-centric) | $-0.311 \pm 0.359$ | $0.242 \pm 0.246$ | $-0.069 \pm 0.404$ |
> > | + offline data ($\mathcal{D}$-centric) | $-0.391 \pm 0.387$ | $0.365 \pm 0.298$ | $-0.026 \pm 0.357$ |
> > | + offline data + reset ($\mathcal{D}$-centric) | $-0.632 \pm 0.349$ | $0.486 \pm 0.340$ | $-0.146 \pm 0.393$ |
> > | + offline RL + offline data (mixed $\pi_0+\mathcal{D}$) | $-0.247 \pm 0.345$ | $0.226 \pm 0.206$ | $-0.021 \pm 0.357$ |
> >
> > ### 500K steps
> > | Fine-tuning method | Stability | Plasticity | Improvement |
> > |-|-|-|-|
> > | baseline | $-0.461 \pm 0.381$ | $0.667 \pm 0.375$ | $0.206 \pm 0.502$ |
> > | + warmup ($\pi_0$-centric) | $-0.453 \pm 0.382$ | $0.689 \pm 0.372$ | $0.236 \pm 0.495$ |
> > | + offline RL ($\pi_0$-centric) | $-0.344 \pm 0.362$ | $0.431 \pm 0.327$ | $0.087 \pm 0.526$ |
> > | + offline data ($\mathcal{D}$-centric) | $-0.442 \pm 0.388$ | $0.745 \pm 0.316$ | $0.303 \pm 0.415$ |
> > | + offline data + reset ($\mathcal{D}$-centric) | $-0.607 \pm 0.360$ | $0.910 \pm 0.299$ | $0.304 \pm 0.318$ |
> > | + offline RL + offline data (mixed $\pi_0+\mathcal{D}$) | $-0.291 \pm 0.360$ | $0.447 \pm 0.307$ | $0.155 \pm 0.433$ |

---

### Official Review · Reviewer_cEvg · 2025-10-31

**Soundness:** 3
**Presentation:** 4
**Contribution:** 3
**Rating:** 8
**Confidence:** 4

**Summary:**

This paper discusses three regimes in offline-to-online RL, and analyses the design decisions in different algorithms to show which ones are suited for which regimes. The three regimes are defined as the difference between the performance of the offline RL policy, and the performance of the data collection policy. Superior regimes (pretrained policy better than dataset) require online policy to stay close to initialization; inferior regimes need to retain the offline dataset, and parameter reset can also be beneficial; comparable regimes yield similar performances.

**Strengths:**

- offers a unique insight into the different regimes of offline-to-online RL, and offers a great discussion for the community into why certain RL algorithms are better than other for certain regimes. This is the first work that I am aware of that classifies different regimes based on the performance different of the pre-trained policy and the data collection policy.
- writing is super clear and clearly explains the different regimes and which methods excel. I appreciate the take away section at the bottom of every results section.
- clear and comprehensive experiments to support the paper’s claims

**Weaknesses:**

- Section 3.2 offers definitions of new concepts such as stability, plasticity, and knowledge decomposition. However, as far as I can tell, these definitions were never used or referenced in the subsequent text (which only used J(pi_0) and J(pi_D) to define the three regimes). Why define these concepts here? How are they useful? Why are they defined this way and not other forms? With no discussion, their definition seems a bit random.
- This paper mainly focuses on off policy RL algorithms for offline-to-online RL, yet, it misses any discussion of one critical part of off policy RL – learning a Q/value function. How does the quality of the Q function change in the three different regimes? And how does the quality of the Q function impact fine-tuning performance? Without discussion of the Q function, the paper drew conclusions only on policy performance (J(pi_0) and J(pi_D)), but this is not a full picture to characterize the three regimes. For example, in the “Inferior regime”, is the reason we need to retain the dataset because of a bad policy initialization, or because the pre-trained Q function is bad (and needed a parameter reset). For example, Fig 4 shows that BC-initialization is really bad unless the dataset is kept: this is probably due to the fact that there was no pre-trained Q function. This is the most unsatisfying part of the paper for me. The paper cites WSRL [1], which offers some discussion on how the quality of the pre-trained Q function impacts fine-tuning performance, and I would love to see more discussion of that in this paper.
- There is a missed opportunity to discuss more on how the different pre-training methods influence fine-tuning results in the three different regimes.

[1] Efficient Online Reinforcement Learning Fine-Tuning Need Not Retain Offline Data

**Questions:**

- In the superior regime, why does plasticity not matter? Shouldn’t plasticity matter in general for all three regimes?

---

> ### Author Response · Authors · 2025-11-25
> **Response to cEvg-1**
>
> We are grateful for your comprehensive review and valuable feedback. We will address each of your comments and concerns in the following responses, as well as in our revised manuscript.
>
> ---
> > **W1**: Section 3.2 offers definitions of new concepts such as stability, plasticity, and knowledge decomposition. However, as far as I can tell, these definitions were never used or referenced in the subsequent text. Why define these concepts here? How are they useful? Why are they defined this way and not other forms?
>
> Summary: The usefulness of these concepts is twofold: (1) they are **conceptually useful**, as they formalize the principle of plasticity and stability and; (2) they are now **practically useful**, as we add the empirical values and find them aligned with intuition. Below are details.
>
>
> (1) We define these concepts in order to formalize the principle of plasticity and stability in the context of offline-to-online RL. This provides the main intuition and explanation for why our framework works. These definitions, connect by the knowledge decomposition, offer a structured way to reason about how prior knowledge interacts with fine-tuning methods and why different design choices behave differently across regimes.
>
> (2) Furthermore, these definitions help us better understand and distinguish different fine-tuning methods. We **report the empirical values** of stability, plasticity, and the resulting improvement over $J^*_{off}$ (given by stability plus plasticity) in our empirical studies, as presented below and in the new Section 5.1 in the updated PDF.
>
> **We observe that the empirical values of stability and plasticity align well with intuition**:
>
> 1. **Stability**: The “offline RL + offline data” method achieves the *strongest stability* across all regimes, while the baseline yields the *lowest stability* (except for “offline data + reset,” which performs a full parameter reset).
> 2. **The strongest plasticity is *regime-dependent***: In terms of plasticity, we find that $\mathcal{D}$-centric methods provide the strongest plasticity in the Inferior regime, since the offline data contain high-quality trajectories that are useful for online fine-tuning, but these methods show lower plasticity in the Superior regime. A similar pattern appears for the “warmup” setting: it offers strong plasticity in the Superior regime but lower plasticity in the Inferior regime.
> 3. **The effect of fine-tuning methods are *regime-specific***. Aggregated results show modest differences across methods, but regime-specific tables reveal clear, method-dependent patterns, highlighting the practical utility of the stability–plasticity principle and the three-regime framework.

---

> ### Author Response · Authors · 2025-11-25
> **Response to cEvg-1(table)**
>
> **Table: Empirical values of Stability and Plasticity in Inferior Regime**
>
> Highest values are bolded (for plasticity, the second-highest is bolded since the reset method is always the highest).
> | Fine-tuning method| Stability| Plasticity| Improvement = Stability + Plasticity |
> |--|--|--|--|
> | baseline| -0.711 ± 0.305| 0.561 ± 0.378| -0.150 ± 0.467|
> | + warmup ($\pi_0$-centric) | -0.692 ± 0.302| 0.577 ± 0.382| -0.116 ± 0.475                         |
> | + offline RL ($\pi_0$-centric)| -0.659 ± 0.319 | 0.297 ± 0.292| -0.362 ± 0.498|
> | + offline data ($\mathcal{D}$-centric)| -0.694 ± 0.304 | **0.849 ± 0.188**| 0.155 ± 0.278|
> | + offline data + reset ($\mathcal{D}$-centric)| -0.769 ± 0.275 | 0.951 ± 0.138| **0.182 ± 0.275**|
> | + offline RL + offline data (mixed $\pi_0+\mathcal{D}$)| **-0.616 ± 0.352** | 0.475 ± 0.284 | -0.142 ± 0.437 |
>
>
> **Table: Empirical values of Stability and Plasticity in Superior Regime**
> | Fine-tuning method| Stability| Plasticity| Improvement = Stability + Plasticity |
> |--|---|--|--|
> | baseline| -0.296 ± 0.329| 0.834 ± 0.275| 0.538 ± 0.271 |
> | + warmup ($\pi_0$-centric)| -0.296 ± 0.336| **0.883 ± 0.242**|**0.587 ± 0.267** |
> | + offline RL  ($\pi_0$-centric)| -0.158 ± 0.250| 0.521 ± 0.292| 0.363 ± 0.261 |
> | + offline data ($\mathcal{D}$-centric)| -0.292 ± 0.346     | 0.820 ± 0.270| 0.528 ± 0.258 |
> | + offline data + reset ($\mathcal{D}$-centric)| -0.467 ± 0.325| 1.026 ± 0.175| 0.559 ± 0.290 |
> | + offline RL + offline data (mixed $\pi_0+\mathcal{D}$)| **-0.104 ± 0.211** | 0.394 ± 0.288| 0.290 ± 0.242 |
>
>
> **Table: Empirical values of Stability and Plasticity in Comparable Regime**
> | Fine-tuning method| Stability| Plasticity| Improvement = Stability + Plasticity |
> |-|--|---|--|
> | baseline| -0.030 ± 0.047| 0.573 ± 0.420| 0.543 ± 0.452 |
> | + warmup ($\pi_0$-centric)| -0.029 ± 0.047| 0.562 ± 0.401| 0.533 ± 0.432 |
> | + offline RL ($\pi_0$-centric)| -0.030 ± 0.047| 0.575 ± 0.387| 0.545 ± 0.414 |
> | + offline data  ($\mathcal{D}$-centric)| -0.029 ± 0.047| 0.694 ± 0.347| 0.664 ± 0.375 |
> | + offline data + reset ($\mathcal{D}$-centric)| -0.087 ± 0.136| 0.674 ± 0.380| 0.587 ± 0.386 |
> | + offline RL + offline data (mixed $\pi_0+\mathcal{D}$)| -0.024 ± 0.037 | 0.692 ± 0.307      | 0.668 ± 0.326 |
>
> **Table: Empirical values of Stability and Plasticity in all regimes**
> | Fine-tuning method| Stability| Plasticity| Improvement = Stability + Plasticity |
> |-|--|---|--|
> | baseline| -0.414 ± 0.384| 0.701 ± 0.361| 0.287 ± 0.503|
> | + warmup ($\pi_0$-centric)| -0.407 ± 0.380| 0.730 ± 0.356| 0.323 ± 0.503|
> | + offline RL ($\pi_0$-centric)| -0.324 ± 0.367         | 0.446 ± 0.326| 0.122 ± 0.533|
> | + offline data  ($\mathcal{D}$-centric)| -0.405 ± 0.387| 0.814 ± 0.259| 0.409 ± 0.344|
> | + offline data + reset  ($\mathcal{D}$-centric)| -0.529 ± 0.362| 0.954 ± 0.230| 0.425 ± 0.350|
> | + offline RL + offline data  (mixed $\pi_0+\mathcal{D}$)| -0.281 ± 0.365| 0.461 ± 0.304| 0.181 ± 0.432|

---

> ### Author Response · Authors · 2025-11-25
> **Response to cEvg-2**
>
> >**W2-1**: How does the quality of the Q function change in the three different regimes?
>
> In the new **Sec. 5.5** and **Figure 6** of the revised manuscript, we examine how the Q-function evolves during fine-tuning in the Superior regime and Inferior regime, using the same metrics in the WSRL paper.
>
> In both regimes, the two methods exhibit similarly low TD loss on the online data distribution. As expected, fine-tuning with the offline dataset results in small TD loss on the offline dataset. However, warmup (without offline dataset) produces **much larger TD loss on the offline dataset** in the Inferior regime than in the Superior regime, especially in the first 100k steps, and the corresponding **Q-values diverge more severely**. This further illustrates why the offline dataset is crucial in the Inferior regime, whereas it is not necessary in the Superior regime.
>
> >**W2-2**: And how does the quality of the Q function impact fine-tuning performance?
>
> In the new **Section C** of the updated PDF, we show an alternative taxonomy based on the Q functions. Its prediction accuracy on fine-tuning results is worse than our performance-based taxonomy. Below is the detail.
>
> Conservative offline RL aims to outperform the behavior policy that generates the offline dataset. Ideally, the value function of the pretrained policy $\pi_0$ should therefore exceed that of the behavior policy $\pi_\mathcal D$ on in-dataset state-action pairs.
> Formally, Lemma 2 of [1] show that for in-sample Q-learning, the optimal value function satisfies
>
> $$Q^{\pi_0^*}(s,a) \ge Q^{\pi_\mathcal D}(s,a), \quad \forall (s,a) \in \mathcal D,$$
>
> where $\pi_0^*$ denotes the optimal pretrained policy and $Q^\pi$ is the ground-truth value function of policy $\pi$. A weaker, expectation-based version of this condition is
>
> $$\mathbb E_{(s,a) \sim \mathcal D}[Q^{\pi_0^*}(s,a)] \ge \mathbb E_{(s,a) \sim \mathcal D}[ Q^{\pi_\mathcal D}(s,a)].$$
>
> Motivated by this result, we examine a taxonomy based on comparing $\mathbb E_{(s,a) \sim \mathcal D}[\hat Q^{\pi_0}(s,a)]$ and $\mathbb E_{(s,a) \sim \mathcal D}[Q^{\pi_\mathcal D}(s,a)]$, where $\hat Q^{\pi_0}$ denotes the pretrained Q-function.
> The intuition is that if $$\mathbb E_{(s,a) \sim \mathcal D}[\hat Q^{\pi_0}(s,a)] \ge \mathbb E_{(s,a) \sim \mathcal D}[Q^{\pi_\mathcal D}(s,a)],$$
> then pretraining approximately satisfies above equation, placing it in the Superior or Comparable regime; otherwise, it falls into the Inferior regime.
>
> In practice, we estimate $Q^{\pi_{\mathcal{D}}}(s,a)$ using the Monte Carlo return $G(s,a)$ from $\mathcal{D}$. We then perform a $t$-test to assess whether $\mathbb{E}_{(s,a) \sim \mathcal D}[\hat{Q}^{\pi_0}(s,a) - G(s,a)]$ is zero. Acceptance of the null corresponds to the Comparable Q-regime; a significantly positive difference indicates the Superior Q-regime, and a significantly negative difference indicates the Inferior Q-regime.
>
> We then evaluate how well this Q-based regime classification aligns with the fine-tuning results. It achieves 32 out of 63 correct predictions (**51%**), 19 opposite mismatches (30%), and 12 adjacent mismatches (19%). While the Q-based regime predictions are much better than random (33%), they are still significantly worse than those produced by our framework (**71%**). The confusion matrix is shown below.
>
> **Table: Confusion matrix on Q-based regimes**
>
> |  | Q-based Superior regime| Q-based Comparable regime| Q-based Inferior regime|
> |--|-|--|-|
> | $\pi_0$-centric $>$ $\mathcal{D}$-centric| **23**| 0| 4|
> | $\pi_0$-centric $\approx$ $\mathcal{D}$-centric| 8| **0**| 3|
> | $\pi_0$-centric $<$ $\mathcal{D}$-centric| 15| 1| **9**|
>
> Reference:
>
> [1] Offline Reinforcement Learning with Implicit Q-Learning. ICLR 2022.
>
> > **W2-3**: For example, in the Inferior regime, is the reason we need to retain the dataset because of a bad policy initialization, or because the pre-trained Q function is bad (and needed a parameter reset).
>
> Since our regime taxonomy is based on the performance of the pretrained agent, a policy initialization is expected to perform poorly in the Inferior regime. Regarding the Q-function, we examined several environments with different pretraining methods that fall into the Superior and Inferior regimes, as shown in the two tables below. However, we do not observe that the TD loss is consistently smaller in the Superior regime compared to the Inferior regime. Therefore, we have not drawn a definitive conclusion about the relationship between Inferior and Q-function.
>
> | Statistic on kitchen-partial-v0| CalQL (Superior)| ReBRAC (Inferior) |
> |--|--|-|
> | $J(\pi_0)$      | $0.764 \pm 0.094$|$0.133 \pm 0.085$|
> | TD loss on offline data  |    $36.0\pm 4.4$       |$2.3\pm0.2$|
>
> | Statistic on halfcheetah-medium-expert-v2| CalQL (Inferior)| ReBRAC (Superior)|
> |--|--|-|
> | $J(\pi_0)$      |$0.519 \pm 0.078$|$1.010 \pm 0.019$|
> | TD loss on offline data  |$98.9\pm 8.9$|$32.0\pm5.8$|

---

> ### Author Response · Authors · 2025-11-25
> **Response to cEvg-3**
>
> > **W2-4**:  For example, Fig 4 shows that BC-initialization is really bad unless the dataset is kept: this is probably due to the fact that there was no pre-trained Q function.
>
> We pretrain the critic using Fitted Q Evaluation (FQE) [1] after BC policy pretraining, so the issue is not due to the absence of a pretrained Q-function.
>
>
> Reference:
>
> [1] Empirical study of off-policy policy evaluation for reinforcement learning. arXiv preprint arXiv:1911.06854
>
>
> ---
>
> >**W3**: There is a missed opportunity to discuss more on how the different pre-training methods influence fine-tuning results in the three different regimes.
>
> In terms of pretraining methods, the offline RL algorithms (CalQL and ReBRAC) are more likely to lead to Superior regimes (14/21 and 13/21, respectively). In contrast, BC pretraining yields the most Comparable and Inferior regimes (16/21).
>
> Here we also provide the confusion matrices for the three pretraining methods. Our framework attains
> - 81% prediction accuracy for CalQL pretraining,
> - 67% for ReBRAC pretraining, and
> - 67% for BC pretraining.
>
> **Table: Confusion matrix for CalQL pretraining**
> | fine-tuning \ regime | superior | comparable | inferior |
> |--|-|--|-|
> | $\pi_0$-centric $>$ $\mathcal{D}$-centric| **11**| 0| 1|
> | $\pi_0$-centric $\approx$ $\mathcal{D}$-centric| 3| **1**| 0|
> | $\pi_0$-centric $<$ $\mathcal{D}$-centric| 0| 0| **5**|
>
> **Table: Confusion matrix for ReBRAC pretraining**
> | fine-tuning \ regime | superior | comparable | inferior |
> |--|-|--|-|
> | $\pi_0$-centric $>$ $\mathcal{D}$-centric| **8**| 0|0|
> | $\pi_0$-centric $\approx$ $\mathcal{D}$-centric| 3| **1**| 1|
> | $\pi_0$-centric $<$ $\mathcal{D}$-centric| 2| 1| **5**|
>
> **Table: Confusion matrix for BC pretraining**
> | fine-tuning \ regime | superior | comparable | inferior |
> |--|-|--|-|
> | $\pi_0$-centric $>$ $\mathcal{D}$-centric| **5**| 2| 0|
> | $\pi_0$-centric $\approx$ $\mathcal{D}$-centric| 0| **0**| 2|
> | $\pi_0$-centric $<$ $\mathcal{D}$-centric| 0| 3| **9**|
>
>
>
> ---
>
> >**Q1**: In the superior regime, why does plasticity not matter? Shouldn’t plasticity matter in general for all three regimes?
>
> Yes, plasticity is relevant in all three regimes. In Figure 2, we emphasized its role in the *Inferior* regime because this regime sometimes requires **very strong plasticity** (e.g., a full parameter reset) to enable efficient fine-tuning. In the *Superior* and *Comparable* regimes, plasticity is also necessary, but typically not to the same extent.
>
> We have revised Figure 2 in the new manuscript by adding dashed arrows between the other two regimes and plasticity to better reflect its relevance across all regimes.
>
>
>
> ---
>
> We hope that our response can address the reviewer’s concerns, and we are happy to provide further details should the reviewer find any aspect insufficiently explained.

---

> > ### Comment · Reviewer_cEvg · 2025-11-26
> >
> > Thanks for providing a detailed response to all my questions. I also appreciate all the new sections and experiments you added during rebuttal, and I think it's really helpful and improves the paper. I will keep my score. I would encourage the authors to expand on these directions (explaining the effectiveness of plasticity/stability metric, alternative taxonomies) even more for the camera ready version.

---

> > > ### Author Response · Authors · 2025-12-03
> > >
> > > Thank you for your encouraging response.
> > >
> > > we identified a minor mistake in the aggregation of the empirical values. We have corrected it and present the corrected results in the following tables. Importantly, this fix does *not* affect our conclusions.
> > >
> > > We have also added transition sentences to the Introduction, Section 3, and Section 5 to make the newly added sections better integrated and more coherent with the rest of the paper.
> > >
> > > ## Corrected Table: Empirical values of Stability and Plasticity in Superior Regime
> > > ### 50K steps
> > > | Fine-tuning method | Stability | Plasticity | Improvement |
> > > |-|-|-|-|
> > > | baseline | $-0.352 \pm 0.344$ | $0.525 \pm 0.325$ | $0.172 \pm 0.145$ |
> > > | + warmup ($\pi_0$-centric) | $-0.328 \pm 0.348$ | $0.501 \pm 0.315$ | $\mathbf{0.173 \pm 0.122}$ |
> > > | + offline RL ($\pi_0$-centric) | $-0.162 \pm 0.242$ | $0.289 \pm 0.255$ | $0.127 \pm 0.151$ |
> > > | + offline data ($\mathcal{D}$-centric) | $-0.293 \pm 0.351$ | $0.448 \pm 0.318$ | $0.155 \pm 0.138$ |
> > > | + offline data + reset ($\mathcal{D}$-centric) | $-0.615 \pm 0.338$ | $\mathbf{0.581 \pm 0.349}$ | $-0.034 \pm 0.311$ |
> > > | + offline RL + offline data (mixed $\pi_0+\mathcal{D}$) | $\mathbf{-0.072 \pm 0.128}$ | $0.204 \pm 0.189$ | $0.132 \pm 0.138$ |
> > >
> > > ### 500K steps
> > > | Fine-tuning method | Stability | Plasticity | Improvement |
> > > |-|-|-|-|
> > > | baseline | $-0.399 \pm 0.363$ | $0.832 \pm 0.275$ | $0.433 \pm 0.276$ |
> > > | + warmup ($\pi_0$-centric) | $-0.394 \pm 0.371$ | $0.865 \pm 0.251$ | $\mathbf{0.471 \pm 0.258}$ |
> > > | + offline RL ($\pi_0$-centric) | $-0.199 \pm 0.268$ | $0.519 \pm 0.291$ | $0.319 \pm 0.256$ |
> > > | + offline data ($\mathcal{D}$-centric) | $-0.376 \pm 0.380$ | $0.796 \pm 0.282$ | $0.421 \pm 0.251$ |
> > > | + offline data + reset ($\mathcal{D}$-centric) | $-0.615 \pm 0.338$ | $\mathbf{1.011 \pm 0.183}$ | $0.396 \pm 0.271$ |
> > > | + offline RL + offline data (mixed $\pi_0+\mathcal{D}$) | $\mathbf{-0.124 \pm 0.213}$ | $0.393 \pm 0.288$ | $0.270 \pm 0.233$ |
> > >
> > > ---
> > >
> > > ## Corrected Table: Empirical values of Stability and Plasticity in Inferior Regime
> > > ### 50K steps
> > > | Fine-tuning method | Stability | Plasticity | Improvement |
> > > |-|-|-|-|
> > > | baseline | $-0.688 \pm 0.348$ | $0.216 \pm 0.219$ | $-0.472 \pm 0.408$ |
> > > | + warmup ($\pi_0$-centric) | $-0.671 \pm 0.343$ | $0.213 \pm 0.217$ | $-0.458 \pm 0.422$ |
> > > | + offline RL ($\pi_0$-centric) | $-0.665 \pm 0.342$ | $0.159 \pm 0.197$ | $-0.505 \pm 0.428$ |
> > > | + offline data ($\mathcal{D}$-centric) | $-0.670 \pm 0.343$ | $0.268 \pm 0.215$ | $-0.402 \pm 0.398$ |
> > > | + offline data + reset ($\mathcal{D}$-centric) | $-0.832 \pm 0.191$ | $\mathbf{0.417 \pm 0.301}$ | $-0.415 \pm 0.395$ |
> > > | + offline RL + offline data (mixed $\pi_0+\mathcal{D}$) | $\mathbf{-0.640 \pm 0.357}$ | $0.266 \pm 0.219$ | $\mathbf{-0.374 \pm 0.415}$ |
> > >
> > > ### 500K steps
> > > | Fine-tuning method | Stability | Plasticity | Improvement |
> > > |-|-|-|-|
> > > | baseline | $-0.692 \pm 0.307$ | $0.475 \pm 0.378$ | $-0.217 \pm 0.480$ |
> > > | + warmup ($\pi_0$-centric) | $-0.677 \pm 0.306$ | $0.495 \pm 0.389$ | $-0.182 \pm 0.490$ |
> > > | + offline RL ($\pi_0$-centric) | $-0.650 \pm 0.322$ | $0.262 \pm 0.283$ | $-0.388 \pm 0.489$ |
> > > | + offline data ($\mathcal{D}$-centric) | $-0.674 \pm 0.308$ | $0.696 \pm 0.340$ | $0.023 \pm 0.451$ |
> > > | + offline data + reset ($\mathcal{D}$-centric) | $-0.769 \pm 0.275$ | $\mathbf{0.870 \pm 0.299}$ | $\mathbf{0.101 \pm 0.205}$ |
> > > | + offline RL + offline data (mixed $\pi_0+\mathcal{D}$) | $\mathbf{-0.614 \pm 0.354}$ | $0.440 \pm 0.297$ | $-0.174 \pm 0.441$ |
> > >
> > > ---
> > >
> > > ## Corrected Table: Empirical values of Stability and Plasticity in Comparable Regime
> > > ### 50K steps
> > > | Fine-tuning method | Stability | Plasticity | Improvement |
> > > |-|-|-|-|
> > > | baseline | $-0.083 \pm 0.084$ | $0.163 \pm 0.237$ | $0.080 \pm 0.251$ |
> > > | + warmup ($\pi_0$-centric) | $-0.082 \pm 0.085$ | $0.163 \pm 0.195$ | $0.081 \pm 0.212$ |
> > > | + offline RL ($\pi_0$-centric) | $-0.079 \pm 0.085$ | $0.214 \pm 0.264$ | $0.135 \pm 0.270$ |
> > > | + offline data ($\mathcal{D}$-centric) | $-0.081 \pm 0.085$ | $0.147 \pm 0.161$ | $0.066 \pm 0.166$ |
> > > | + offline data + reset ($\mathcal{D}$-centric) | $-0.114 \pm 0.147$ | $\mathbf{0.261 \pm 0.250}$ | $0.147 \pm 0.237$ |
> > > | + offline RL + offline data (mixed $\pi_0+\mathcal{D}$) | $\mathbf{-0.057 \pm 0.058}$ | $0.240 \pm 0.251$ | $\mathbf{0.182 \pm 0.258}$ |
> > >
> > > ### 500K steps
> > > | Fine-tuning method | Stability | Plasticity | Improvement |
> > > |-|-|-|-|
> > > | baseline | $-0.043 \pm 0.071$ | $0.560 \pm 0.411$ | $0.516 \pm 0.452$ |
> > > | + warmup ($\pi_0$-centric) | $-0.042 \pm 0.072$ | $0.543 \pm 0.390$ | $0.501 \pm 0.423$ |
> > > | + offline RL ($\pi_0$-centric) | $-0.043 \pm 0.072$ | $0.564 \pm 0.382$ | $0.521 \pm 0.416$ |
> > > | + offline data ($\mathcal{D}$-centric) | $-0.042 \pm 0.072$ | $0.680 \pm 0.338$ | $0.638 \pm 0.361$ |
> > > | + offline data + reset ($\mathcal{D}$-centric) | $-0.106 \pm 0.139$ | $0.623 \pm 0.431$ | $0.516 \pm 0.431$ |
> > > | + offline RL + offline data (mixed $\pi_0+\mathcal{D}$) | $\mathbf{-0.035 \pm 0.061}$ | $\mathbf{0.682 \pm 0.302}$ | $\mathbf{0.646 \pm 0.314}$ |

---

> > > > ### Author Response · Authors · 2025-12-03
> > > >
> > > > ## Corrected Table: Empirical values of Stability and Plasticity in all regimes
> > > >
> > > > ### 50K steps
> > > > | Fine-tuning method | Stability | Plasticity | Improvement |
> > > > |-|-|-|-|
> > > > | baseline | $-0.433 \pm 0.379$ | $0.397 \pm 0.329$ | $-0.036 \pm 0.395$ |
> > > > | + warmup ($\pi_0$-centric) | $-0.413 \pm 0.380$ | $0.382 \pm 0.315$ | $-0.031 \pm 0.389$ |
> > > > | + offline RL ($\pi_0$-centric) | $-0.311 \pm 0.359$ | $0.242 \pm 0.246$ | $-0.069 \pm 0.404$ |
> > > > | + offline data ($\mathcal{D}$-centric) | $-0.391 \pm 0.387$ | $0.365 \pm 0.298$ | $-0.026 \pm 0.357$ |
> > > > | + offline data + reset ($\mathcal{D}$-centric) | $-0.632 \pm 0.349$ | $0.486 \pm 0.340$ | $-0.146 \pm 0.393$ |
> > > > | + offline RL + offline data (mixed $\pi_0+\mathcal{D}$) | $-0.247 \pm 0.345$ | $0.226 \pm 0.206$ | $-0.021 \pm 0.357$ |
> > > >
> > > > ### 500K steps
> > > > | Fine-tuning method | Stability | Plasticity | Improvement |
> > > > |-|-|-|-|
> > > > | baseline | $-0.461 \pm 0.381$ | $0.667 \pm 0.375$ | $0.206 \pm 0.502$ |
> > > > | + warmup ($\pi_0$-centric) | $-0.453 \pm 0.382$ | $0.689 \pm 0.372$ | $0.236 \pm 0.495$ |
> > > > | + offline RL ($\pi_0$-centric) | $-0.344 \pm 0.362$ | $0.431 \pm 0.327$ | $0.087 \pm 0.526$ |
> > > > | + offline data ($\mathcal{D}$-centric) | $-0.442 \pm 0.388$ | $0.745 \pm 0.316$ | $0.303 \pm 0.415$ |
> > > > | + offline data + reset ($\mathcal{D}$-centric) | $-0.607 \pm 0.360$ | $0.910 \pm 0.299$ | $0.304 \pm 0.318$ |
> > > > | + offline RL + offline data (mixed $\pi_0+\mathcal{D}$) | $-0.291 \pm 0.360$ | $0.447 \pm 0.307$ | $0.155 \pm 0.433$ |

---

### Official Review · Reviewer_6yqr · 2025-11-01

**Soundness:** 2
**Presentation:** 4
**Contribution:** 1
**Rating:** 2
**Confidence:** 4

**Summary:**

This paper introduces an empirical framework to predict which kind of offline-to-online RL algorithm will perform better for a given dataset and policy, based on the relative performances of the data-collection policy and the given policy we will fine-tune. The proposed takeaways are that if the given policy performs better than the data-collection policy, methods that provide stability around the pre-trained policy will perform better, whereas if the data-collection policy is better, methods that provide stability around the dataset will be better.

**Strengths:**

- The authors provide very thorough empirical results. They test 21 dataset-task tuples, and 3 pre-training methods per tuple.
- The writing is generally clear and focused.
- Code is released, which is appreciated to support reproducibility.

**Weaknesses:**

I have key concerns regarding the definitions of the family of methods and the interpretations of the results, which leaves me unconvinced about the central claims in the paper.

**Key concern**: why does offline RL regularization during fine-tuning classify as stability with respect to the pre-trained policy $\pi_0$? If my understanding is correct, the same method used for the offline phase is used during the online phase for offline RL regularization. So for example, in figures 3, 4, 5, if CalQL was used for pretraining, then the same CalQL coefficient will be used for online fine-tuning. Why do the authors interpret CalQL as providing stability with respect to the pretrained policy $\pi_0$? It does not keep a frozen copy of this policy at all. I would actually argue it provides stability relative to the offline dataset $D$, since it increases the Q-values of state-action tuples in the dataset, and decreases for on-policy and random action samples. For BC-pretrained policies, does offline RL mean TD3+BC in this paper? If so, the same argument applies - this does not provide stability w.r.t. the pretrained policy. With correct labeling of methods, conclusions in the paper might change significantly.


**Clarity / other concerns:**
- Line 216 Minimal baseline is introduced, but it is not explained? It is then mentioned again in line 268, but it’s still unclear (“minimal baseline corresponds to maximum plasticity with no explicit stability mechanism”, so SAC?)
- Line 258: I don’t think this perspective on RLPD is meaningful, since there is no pre-training at all. This should be rephrased to make this clear (right now might read as “some networks are kept, while others are reset for plasticity”).
- Results on figure 3: offline RL + offline data should be equivalent to CalQL right? Why is performance significantly worse than the reported numbers in the CalQL paper? E.g. halfcheetah-medium-v2, from a CalQL init (top middle plot), the green line plateaus at 60%, but CalQL reports 93% after fine-tuning. Likewise with hopper-random-v2, and to a lesser extent antmaze-large-diverse-v2.
- Line 356:  “This aggregate outcome strongly supports our principle that, in the superior regime, …” I don’t see strong support from figure 3. The green method is a method that tries to stick close to dataset behaviors, and it is the best performing-method on the top left figure. The top right figure shows large overlapping confidence intervals. Same with walker2d-medium-replay-v2, BC.
- The methods compared are not explained properly. E.g. on figure 3 for “walker2d-medium-replay-v2, BC”, it is clear that you pre-train the policy with BC. However, how do you apply offline RL? Do you pre-train a critic, or start the critic from scratch for the online phase? If so, then the orange line (offline RL without offline data) is not meaningful. What type of offline RL regularization do you apply for the orange and green lines?
- In figure 5, it doesn’t look like $\pi_0 \approx D$. The green line, which is a method that tries to stay close to the support of the dataset (i.e. a stability relative to the offline dataset approach) seems to perform better than other methods.

**Questions:**

Eqn 4: shouldn’t it be J_off^* instead of min J(pi_j)? Otherwise this doesn’t measure how much we can improve during fine-tuning, but rather the range of values during fine-tuning, which might be much larger if stability is low.

---

> ### Author Response · Authors · 2025-11-25
> **Response to 6yqr-1**
>
> We are grateful for your comprehensive review and valuable feedback. We will address each of your comments and concerns in the following responses, as well as in our revised manuscript.
>
> ---
> **Key concern**:
> > **W1**: why does offline RL regularization during fine-tuning classify as stability with respect to the pre-trained policy $\pi_0$? ... It does not keep a frozen copy of this policy at all. I would actually argue it provides stability relative to the offline dataset, since it increases the Q-values of state-action tuples in the dataset, and decreases for on-policy and random action samples.
>
> We summarize your key concern as: During fine-tuning,
> - *Why is CalQL considered providing stability w.r.t. the pretrained policy $\pi_0$, given $\pi_0$ is not stored?*
> - *Why is CalQL not considered as stability w.r.t. offline dataset $\mathcal D$ since the values of in-dataset state-actions are increased?*
>
> We believe the answers to both questions depend on **whether we replay the offline dataset $\mathcal D$ during fine-tuning**. Below we discuss the two scenarios.
>
> **1. Fine-tuning without the offline dataset $\mathcal D$.** In this scenario, as you mentioned, the CalQL regularization increases the Q-values of the state–action tuples in the dataset, i.e., the **online replay buffer $\mathcal B$**. The online buffer $\mathcal B$ is collected by a sequence of policies that originate from $\pi_0$ and evolve to $\pi_N$ during fine-tuning. Thus, offline RL regularization is *more* related to stability w.r.t. $\pi_0$ rather than $\mathcal D$ which is discarded during fine-tuning.
>
>
> **2. Fine-tuning with the offline dataset $\mathcal D$.** In this scenario, it is true that the regularization increases the Q-values of the tuples in offline dataset $D$ *as well as* those from online replay buffer $\mathcal B$. In this case, we labeled the method as “mixed $\pi_0$ + $D$” method in our work.
>
> In summary, even though we do not store $\pi_0$ during online fine-tuning, **offline RL regularization provides stability to $\pi_0$ in an implicit way**. With offline data replay, offline RL regularization also directly provide stability to $\mathcal D$.
>
> > **W1** (continued): For BC-pretrained policies, does offline RL mean TD3+BC in this paper? If so, the same argument applies - this does not provide stability w.r.t. the pretrained policy.
>
> For BC-pretrained policies, we use **ReBRAC** as the offline RL method during fine-tuning. In the updated PDF, we have revised the main paper to make this clearer.
>
> ReBRAC contains two regularization terms: the actor regularization is a BC loss: $-\beta_1 \cdot \bigl(\pi(s) - a\bigr)^2$. The critic regularization is $-\beta_2 \cdot (a' - \hat{a}')^2$, where $a'$ is $\pi(s')$ and $\hat{a}'$ is the next action in the dataset.
>
> Similar to our previous analysis on CalQL, when applying ReBRAC **without** the offline dataset $D$, ReBRAC make the learning closer to the policies that generated the online replay buffer $\mathcal B$, which originate from $\pi_0$ and evolve during fine-tuning.
>
> We have revised the Section 4.2 to make it clearer.
>
> References:
>
> [1] Cal-ql: Calibrated offline rl pre-training for efficient online finetuning. NeurIPS 2023.
>
> [2] Revisiting the Minimalist Approach to Offline Reinforcement Learning. NeurIPS 2023.
>
> ---
>
> **Clarity / other concerns**:
> > **W2**: Minimal baseline is unclear ... so SAC?
>
> The Minimal baseline applies a standard online RL algorithm (e.g., SAC or TD3) initialized with an offline-pretrained agent. We have revised the manucript to make it clearer.
>
>
> ---
>
> >**W3**: I don’t think this perspective on RLPD is meaningful, since there is no pre-training at all. This should be rephrased to make this clear (right now might read as “some networks are kept, while others are reset for plasticity”).
>
> We interpret RLPD within our framework as “pretraining followed by a *fully parameter reset*,” because our goal is to view RLPD as a type of offline-to-online method under a unified perspective. We have revised this to address the *fully parameter reset* to make it clearer.

---

> ### Author Response · Authors · 2025-11-25
> **Response to 6yqr-2**
>
> ---
>
> >**W4**: Figure 3, offline RL + offline data should be equivalent to CalQL, right? Some CalQL results are significantly worse than the reported numbers in the CalQL paper.
>
> While our “offline RL + offline data” setting (with CalQL pretraining) is *conceptually aligned* with applying CalQL during online fine-tuning, the two are *not identical in practice*.
>
> The [official CalQL codebase](https://github.com/nakamotoo/Cal-QL) provides implementations and hyperparameters only for AntMaze and Adroit, and the CalQL paper [1] does not report the hyperparameters for CalQL on the MuJoCo locomotion tasks (see the [github issue](https://github.com/nakamotoo/Cal-QL/issues/5)). Consequently, in our empirical studies on CalQL, we base on the [WSRL's codebase](https://github.com/zhouzypaul/wsrl), which provides a fully supported implementation across all domains and tasks. This implementation difference may account for the performance discrepancies, particularly the large gap in the two MuJoCo results you mentioned. We include hyperparameter details in the updated PDF.
>
>
> References:
>
> [1] Cal-ql: Calibrated offline rl pre-training for efficient online finetuning. NeurIPS 2023.
>
> ---
>
> >**W5**: Line 356: “This aggregate outcome strongly supports our principle that, in the superior regime, …” I don’t see strong support from figure 3. The green method is a method that tries to stick close to dataset behaviors, and it is the best performing-method on the top left figure. The top right figure shows large overlapping confidence intervals. Same with walker2d-medium-replay-v2, BC.
>
> In the top-left figure, we label this figure as $\pi_0 > \mathcal{D}$ because the $\pi_0$-centric method (purple line) outperforms the $\mathcal{D}$-centric method (brown line). While the green line (“offline RL + offline data,” labeled as mixed $\pi_0 + \mathcal{D}$) performs the best, it is neither a $\pi_0$-centric method nor a $\mathcal{D}$-centric method, and therefore does not impact our conclusion.
>
>
>
> Regarding the two figures with large overlapping confidence intervals, the $t$-test results conclude “better” rather than “similar” because our analysis is based on the last 10 evaluation results (last 50k steps) during online fine-tuning, instead of a single evaluation point. We choose this because method performance can fluctuate, and multiple evaluations give more stable results.
>
> ---
>
> >**W6**: In BC, how do you apply offline RL? Do you pre-train a critic, or start the critic from scratch for the online phase? If so, then the orange line (offline RL without offline data) is not meaningful. What type of offline RL regularization do you apply for the orange and green lines?
>
> We agree that the presentation should be clarified. We have revised the manuscript to more clearly explain how we perform critic pretraining and apply offline RL regularization in the BC pretraining setting.
>
> In our experiments, the online and offline phases of BC are structured as follows:
>
> 1. **BC Pre-training (Actor):**
>    We first train the policy using BC on the offline dataset.
> 2. **Fitted Q Evaluation (Critic):**
>    As noted in Appendix's B.1, after BC training, we train a critic via **Fitted Q Evaluation (FQE)** [2] on the same offline dataset. Thus, when entering the online fine-tuning phase, both the actor (BC) and critic (FQE) have been pre-initialized.
> 3. **Online Fine-Tuning:**
>    For the online RL baseline, we use TD3, initialized with the BC+FQE networks. For the offline RL regularization, we use ReBRAC.
>
> **Clarification of the Orange and Green Curves.** With the above setup, all the BC curves use the BC+FQE initialization for online fine-tuning. And we use ReBRAC to the Orange (offline RL) and Green lines (offline RL+offline data).
>
> Reference:
>
> [2] Empirical study of off-policy policy evaluation for reinforcement learning. arXiv preprint arXiv:1911.06854
>
> >**W7**: In figure 5, it doesn’t look like $\pi \approx D$. The green line, which is a method that tries to stay close to the support of the dataset (i.e. a stability relative to the offline dataset approach) seems to perform better than other methods.
>
> The statement $\pi \approx \mathcal{D}$ in this context is only meant to compare the performance of $\pi_0$-centric methods versus $\mathcal{D}$-centric methods.  While the green line (“offline RL + offline data”, labeled as mixed $\pi_0 + \mathcal{D}$) performs the best in the second figure, it does not impact our conclusion.
>
> ---

---

> ### Author Response · Authors · 2025-11-25
> **Response to 6yqr-3**
>
> > **Q1**: Equation 4 (definition of plasticity): shouldn’t it be $J_{off}^*$ instead of min $J(\pi_j)$?
>
> (1) In our definition, plasticity itself does not measure how much improvement can be achieved during fine-tuning. According to our performance decomposition, the improvement over the prior knowledge $J_{off}^*$ is given by the sum of plasticity and stability. This indicates that achieving greater improvement requires taking both plasticity and stability into account.
>
> (2) If $J_{off}^*$ were used in place of $\min_j J(\pi_j)$, the resulting knowledge decomposition would fail to incorporate stability, and therefore would not explicitly capture the trade-off between stability and plasticity.
>
> ---
>
>
> We hope that our responses can address the reviewer’s concerns, and we are happy to provide further details should the reviewer find any aspect insufficiently explained.

---

### Official Review · Reviewer_gqSs · 2025-11-04

**Soundness:** 3
**Presentation:** 3
**Contribution:** 3
**Rating:** 6
**Confidence:** 5

**Summary:**

This paper introduces the stability-plasticity principle to explain the inconsistent empirical behavior observed in offline-to-online Reinforcement Learning. The authors propose a stability-plasticity principle for offline-to-online RL and a taxonomy of three regimes: Superior, Comparable, and Inferior. For each regime, the paper conducted an extensive empirical study and prescribed different fine-tuning tactics, either prioritizing stability around the offline pretrained policy or stability around the offline dataset, or a mixture of both.  A large-scale empirical study is conducted to validate the framework, finding that the results align closely with the predicted regime-specific prescriptions

**Strengths:**

- The paper is well written and the stability–plasticity principle provides a unified explanation for previously conflicting results in offline-to-online RL.

- It clearly categorizes practical algorithmic choices (warm-up, replay, regularization, reset, etc.) within the framework.

- The empirical evaluation is extensive and supports the paper’s recommendations.

**Weaknesses:**

- The paper provides an empirical analysis of best practices across offline-to-online regimes, but it lacks a unified algorithm that automatically infers the current regime and selects the appropriate design choices. I recommend adding a simple, practical regime-detection algorithm that practitioners can use.

- It is unclear whether approximating the dataset knowledge $J(\pi_D)$ by the dataset’s average accumulated return is appropriate. Why not use the behavioral cloning performance as an alternative proxy for $J(\pi_D)$)?

- The experiments are mostly carried out on D4RL state base tasks. It will be great to further test it out on pixel-based tasks to see if the same framework holds true on high-dimensional settings as well.

**Questions:**

- Since the paper notes that BC typically achieves performance comparable to the dataset itself, why are so many BC-pretrained settings classified as Inferior or Superior?

- How does your framework relate to dataset coverage? Offline RL usually does well in high-coverage regimes and poorly in narrow-data regimes, does this map onto the stability-plasticity explanation?

---

> ### Author Response · Authors · 2025-11-25
> **Response to gqSs-1**
>
> We are grateful for your comprehensive review and valuable feedback. We will address each of your comments and concerns in the following responses, as well as in our revised manuscript.
>
> ---
>
> > **W1**: The paper provides an empirical analysis of best practices across offline-to-online regimes, but it lacks a unified algorithm that automatically infers the current regime and selects the appropriate design choices. I recommend adding a simple, practical regime-detection algorithm that practitioners can use.
>
> In the paper, we provide a unified algorithm to automatically  (1) **infer the current regime** and then (2) **select the design choice family** ($\pi_0$-centric or $\mathcal D$-centric). Below are details.
>
> **(1) We have an algorithm to infer the current regime.** As described in Sec. 5 and appendix B.1, we employ the $t$-test procedure with a margin of $\delta = 0.05$ and a significance level of $\alpha = 0.05$ to infer the current regime. The goal is to formally assess whether the pretrained policy $\pi_0$ and the offline dataset $\pi_{\mathcal{D}}$ are statistically indistinguishable in performance, or whether one is significantly superior. We set a fix $\delta$ since all returns are normalized to roughly lie between 0 and 1. We evaluate three choices of $\delta$ (0.0, 0.05, and 0.1) and select the best-performing value ($\delta=0.05$). The results with $\delta = 0.0$ and $\delta = 0.1$ in Table 3 and Table 4 of the appendix (Table 2 and Table 3 of the initial version).
>
>
> **(2) We have an algorithm to select the $\mathcal D$-centric methods or $\pi_0$-centric methods, but not the single design choice.** We recommend using $\pi_0$-centric methods in the Superior regime and $\mathcal D$-centric methods in the Inferior regime based on the stability–plasticity principle. This already narrows down the design space substantially, making the selection process practical. However, as shown in our empirical studies, no single design choice consistently outperforms the others. Therefore, even with the regime identified, practitioners seeking the best performance may still need to evaluate several different methods within $\mathcal D$-centric methods or $\pi_0$-centric methods and tune their hyperparameters.
>
>
> ---
>
> > **W2**: It is unclear whether approximating the dataset knowledge $J(\pi_D)$ by the dataset’s average accumulated return is appropriate. Why not use the behavioral cloning performance as an alternative proxy for $J(\pi_D)$?
>
> We use the average return of the dataset because it is very easy to obtain and provides useful information. Using the performance of a BC policy as the proxy would be more difficult, since the BC performance can vary significantly depending on implementation details and hyperparameters, and it also requires additional effort to train a BC agent.
>
> Furthermore, using the BC pretraining results, we evaluate how the **BC–based regimes** align with the fine-tuning outcomes. BC-based regimes achieve **41% prediction accuracy** (26/63). The corresponding confusion matrix is presented below. Although the BC-based regimes are better than random (33%), it is significantly worse than our $J(\pi_\mathcal D)$-based framework (**71%**). The results are included in the updated PDF.
>
> **Table: confusion matrix for BC-based regimes**
>
> | | BC-based Superior regime | BC-based Comparable regime | BC-based Inferior regime |
> |--|-|--|-|
> | $\pi_0$-centric $>$ $\mathcal{D}$-centric| **17**| 10| 0|
> | $\pi_0$-centric $\approx$ $\mathcal{D}$-centric| 4| **6**| 1|
> | $\pi_0$-centric $<$ $\mathcal{D}$-centric| 3| 19| **3**|
>
> ---
>
> >**W3**: It will be great to further test it out on pixel-based tasks to see if the same framework holds true on high-dimensional settings as well.
>
> Thank you for your suggestion. We are conducting experiments on pixel-based tasks and will report the results when we have them.

---

> ### Author Response · Authors · 2025-11-25
> **Response to gqSs-2**
>
> > **Q1**: Since the paper notes that BC typically achieves performance comparable to the dataset itself, why are so many BC-pretrained settings classified as Inferior or Superior?
>
> This argument was originally motivated by a common insight from supervised learning, but it may not accurately apply to BC. To avoid potential confusion, we have removed the sentence "BC typically achieves performance comparable to the dataset itself".
>
> Here are our explanation for BC entering Inferior or Superior regime.
>
> **BC entering Inferior regime.** It is possible for BC to enter the *Inferior* regime, because small error in BC lead the agent to  out-of-distribution states, and thus the error acculumates and eventually can significantly degrade its performance. This is known as the *compounding error* issue in BC [1][2][3].
>
> **BC entering Superior regime.** We find that this is mainly due to *imbalanced sampling* over varied-length trajectories. In environments where episodes terminate early when failure happens, e.g., *Hopper* and *Walker2d*, low-return trajectories are typically very short, whereas high-return trajectories are much longer. Because BC is applied in a supervised learning manner, it is updated more frequently on *longer* trajectories, and these trajectories correspond to *higher-return* behavior. As a result, BC can achieve performance that exceeds the average return of the dataset.
>
>
> References:
>
> [1] A Reduction of Imitation Learning and Structured Prediction to No-Regret Online Learning. AISTATS 2011.
>
> [2] Error Bounds of Imitating Policies and Environments. NeurIPS 2020.
>
> [3] Figure 4, Survival Instinct in Offline Reinforcement Learning. NeurIPS 2023.
>
> ---
>
> >**Q2**: How does your framework relate to dataset coverage? Offline RL usually does well in high-coverage regimes and poorly in narrow-data regimes, does this map onto the stability-plasticity explanation?
>
> We answer this question by (1) justifying why coverage is not used in our framework, (2) explaining how coverage relates to our regimes, and (3) discussing the connection between coverage and stability.
>
> (1) **The difficulty of measuring data coverage.** We acknowledge that the value of a dataset in offline RL is not determined solely by its return; coverage is also an important factor. However, it is difficult to quantify a *principled* notion of dataset coverage.
>
> In theory, the data coverage is typically measured by *concentrability coefficient*, for example in [4] (Definition 1), it is defined as
> $$
> \sup_{(s,a)\in \mathcal S\times \mathcal A} \frac{d^\pi(s,a)}{\mu(s,a)},
> $$
> where $\mu$ is behavior policy's state-action distribution and $d^\pi$ is a target policy $\pi$ (often the optimal policy $\pi^*$)'s state-action distribution. It is difficult to estimate this quantity in practice since it requires (1) access to an optimal policy and (2) density-ratio estimation over the whole state-action space.
>
> Therefore, we focus on return as the measurable aspect of dataset usefulness, and we acknowledge limited treatment of coverage as a limitation and an avenue for future work. Nevertheless, we do have some intuition on the role of coverage on the regimes and stability, explained below.
>
> (2) **Connection between coverage and our regimes.** As you noted, offline RL generally favors *high-coverage* regimes, which we expect to correspond more frequently to the *Superior* regime. In contrast, *narrow-data* regimes, such as expert-level datasets, are much more likely to fall into the *Inferior* regime because they offer high-performance but limited-coverage trajectories.
>
> (3) **Connection between coverage and stability.** A *narrow* dataset may provide *less stability* during fine-tuning compared to a high-coverage one. Since the degradation that occurs at the start of fine-tuning is mainly caused by distribution shift, a high-coverage dataset can better anchor the agent’s behavior and reduce this instability, whereas a narrow dataset offers much weaker support.
>
>
> Reference:
>
> [4] Bridging Offline Reinforcement Learning and Imitation Learning: A Tale of Pessimism. NeurIPS 2021.
>
>
> ---
>
> We hope that our response can address the reviewer’s concerns, and we are happy to provide further details should the reviewer find any aspect insufficiently explained.

---

### Author Response · Authors · 2025-11-25
**Common Response**

## Summary of updates in the revised PDF

We highlight the changes in the revised PDF in **blue**. Below is a summary:

- **Section 3.1 (FHZW)**: clarify that we do not assume access to additional quantities such as data coverage and dense-reward proxies.
- **Section 4 (6yqr)**: clarify the minimal baseline.
- **Section 4.2 (6yqr)**: clarify that offline RL regularization implicitly promotes stability around $\pi_0$.
- **Section 5 (6yqr, cEvg)**: clarify that we use FQE to obtain a pretrained critic for BC.
- **New Section 5.1 and appendix B.2 (FHZW, cEvg)**: add the empirical values of stability and plasticity in all settings.
- **New Section 5.5 (cEvg)**: add an analysis of Q-function during fine-tuning.
- **Section 6 (FHZW)**: clarify the limitation when return fails to capture the usefulness of policy and dataset.
- **New appendix B.3 (FHZW)**: provide hyperparameter details.
- **New appendix C (gqSs, cEvg)**: add two alternative taxonomies on regimes.

---

### Comment · Area_Chair_9q9z · 2025-11-28
**Please Check the Authors' Responses**

Dear Reviewers,

The authors have posted their responses. Could you please take a moment to review their responses and check whether your concerns have been adequately addressed (if you have not done it yet)? If possible, kindly initiate the discussion at your earliest convenience.

Your timely assistance is essential for keeping the review process on track. Thank you very much for your support and contribution.

Best regards, Your AC

---

### Author Response · Authors · 2025-12-03
**Summary to AC**

We thank the reviewers for their insightful feedback. Below we summarize the contributions and how we addressed the key concerns.

## Contribution Summary

Our main contribution is to propose the three-regime taxonomy, and a stability–plasticity principle  in offline-to-online RL. This framework not only reconciles previously conflicting empirical findings in the literature but also reveals that best practices are **regime-specific**. In the Superior regime—where the pretrained policy $\pi_0$ outperforms the offline dataset $\mathcal{D}$—$\pi_0$-centric methods are typically more effective. In the Inferior regime—where $\mathcal{D}$ outperforms $\pi_0$—$\mathcal{D}$-centric methods are generally preferable. In the Comparable regime, preserving either source of knowledge leads to similar performance, though outcomes can be sensitive to implementation details.

Our work provides a **unified explanation** for why different offline-to-online RL algorithms succeed in different settings (cEvg, gqSs, FHZW). Our taxonomy offers **the first systematic characterization** based on the relative performances of the pretrained policy $\pi_0$ and the offline dataset $\mathcal{D}$ (cEvg, gqSs). Our **extensive empirical study** further validates the regime-specific predictions across diverse tasks (6yqr, FHZW). Overall, our framework delivers **practical and actionable guidance** for selecting effective offline-to-online RL methods, replacing one-size-fits-all tuning with principled, regime-aware decision-making (cEvg, gqSs, FHZW).


## Rebuttal Summary

The **key concerns** raised by reviewers focused on (1) the usefulness of the definitions of stability and plasticity and the absence of their empirical values, (2) the missing discussion of the Q-function, and (3) the need for additional clarification.

We addressed these concerns as follows:

1.  **Empirical characterization:** We now provide the empirical values of stability, plasticity, and improvement across regimes. These results align well with intuition and further demonstrate the usefulness of our definitions of stability and plasticity.
2. **Q-function analysis**: We added an analysis of the Q-function dynamics during fine-tuning in different regimes.
3. **Alternative taxonomies**: We introduced two additional regime taxonomies—a Q-function–based regime and a BC-based regime. Empirically, our proposed framework achieves the highest prediction accuracy (**71%**), whereas the two alternatives reach only **51%** and **41%**, respectively.
4. **Clarification improvements:** We revised multiple explanations and added targeted clarifying sentences to improve readability and directly address the reviewers’ concerns.

---

### Meta-Review · Area_Chair_qiPD · 2026-01-06

**Summary:**

In this paper, the authors consider the problem of offline-to-online learning combined with offline pre-training, and provided an interesting study on this subject. They classfied the learning situations into three regimes based on the relative performance of the pretrained policy and the average return in the dataset:  Superior, inferior and comparable regimes. The message of the paper is very simple. In the superior regime, where the offline pretrained policy performs better than the average return of the offline dataset, the online fine-tuing phase should not degrade the performance of the initial pretained policy, so-called "stability" w.r.t. the pretrained policy should be emphasized and such online methods should be chosen. On the other hand, in the inferior regime, where the offline pretrained policy performs worse than the average return of the dataset, (in fact this case should be considered as an irrelevant case, such offline algorithms are bad algorithms not even achieving the dataset performance, and should be discarded), on-line algorithms emphasizing the offline dataset such as policy parameter reset are prefered. In the comparable case, both methods can be applied. Then, the authors showed experimental results to validate their argument.

**Reviewer Concerns:**

The major concerns of the reviewers can be summarized as follows:

-  Reviewers gqSs,  FHZW, cEvg: Regime classification for the tasks (Superior vs. Comparable vs. Inferior) is too simplistic and may be subtly incorrect in many scenarios. The key concern here is "is the return performance of just the pretrained policy a correct fully-informative metric to identify the online performance?"  For example, in sparse reward tasks where it is difficult to pre-train a policy to achieve non-trivial success rate directly, but the policy can progress in the task far enough to provide good exploratory data. This is where stability should be preferred over plasticity, but can be overlooked if only the returns are being examined.  Relying the classification entirely on the expected returns is a bit too simplistic for  more challenging and complex tasks.  Reviewer gqSs pointed out that the experiments are mostly carried out on D4RL state base tasks. In the same veain, Reviewer cEvg raised concerns about investigation of Q-function.

- Reviewer FHZW:  The sensitivity of the metrics. The performance can fluctuate so the direct minimum over timesteps may not be a good metric.

- Reviewer 6yqr: Issues on dataset-emphasizing methods or pretrained-policy-emphasizing methods.

**Reviewer Scores:**

Reviewer 6yqr: 2,   Reviewer FHZW: 4, Reviewer gqSs: 6, Reviewer cEvg: 8

It is likely that Reviewers gqSs and cEvg keep their scores as 6 and 8. But, it is also likely that Reviewers 6yqr and FHZW keep their scores as 2 and 4. Indeed, Reviewers 6yqr and FHZW have valid concerns regarding the the paper. The paper provides an interesting study on regrime classification of on-line finetuning combined with offline learning. But, in its current form, the metric seems too simplistic to capture diverse interaction mechanism between offline learning and on-line finetuning. It is recommended that the authors reconsider the formulation,  improve the paper including more informative metric and experiments on more challenging tasks and submit to a future conference.

---

### Decision · Program_Chairs · 2026-01-26

Reject